# Faster Sampling from Gibbs Distributions with Quantum Variance Reduction

## Abstract

We present quantum algorithms that provide provable speedups for approximate sampling from probability distributions of the form $\pi \propto e^{-f}$, where $f$ is a potential function that can be written as a finite sum, i.e., $f = \frac{1}{n} \sum_{i=1}^{n} f_i$. Our approach focuses on stochastic gradient–based methods with only oracle access to individual gradients $\{\nabla f_i\}_{i \in [n]}$. The techniques of our quantum algorithm are based on a non-trivial integration of quantum mean estimation techniques and existing variance reduction techniques such SVRG and CV.

As these techniques often require occasional full-gradient calculations, the key challenge is that an unbalanced weighting between variance reduction and quantum mean estimation results in a regime where the quantum advantage is lost due to frequent full-gradient computation. We overcome this difficulty by carefully optimizing the target variance level. Our algorithms improve the number of gradient queries of classical samplers, such as Hamiltonian Monte Carlo (HMC) and Langevin Monte Carlo (LMC), in terms of dimension, precision, and other problem-dependent parameters.

## 1 Introduction

Efficient sampling from complex distributions is a fundamental problem in many scientific and engineering disciplines, becoming increasingly important as modern applications deal with high-dimensional data and complex probabilistic models. For example, in statistical mechanics, sampling is used to analyze the thermodynamic properties of materials by exploring configurations of particle systems Chandler (1987); Frenkel & Smit (2002). In convex geometry, it helps in approximating volumes and studying high-dimensional structures Lovász & Vempala (2006); Cousins & Vempala (2018). In probabilistic machine learning, sampling plays an important role in Bayesian inference, as it facilitates posterior estimation and quantifies uncertainty in model predictions Welling & Teh (2011); Wang et al. (2015); Durmus & Moulines (2018); Roy et al. (2021). Similarly, in non-convex optimization, sampling allows for the exploration of complex energy landscapes and helps avoid local minima, facilitating progress in tasks such as resource allocation, scheduling, and hyperparameter tuning in machine learning Zhang et al. (2017); Chen et al. (2020).

Given a potential function $f : \mathbb{R}^d \to \mathbb{R}$, we consider the problem of sampling from a probability distribution $\pi$ of the form

$$\pi(\mathbf{x}) = \frac{e^{-f(\mathbf{x})}}{\int e^{-f(\mathbf{x})} \mathrm{d}\mathbf{x}}. \tag{1}$$

This distribution is called the Boltzmann-Gibbs distribution, and our goal is to efficiently sample approximately from $\pi$ while minimizing the number of gradient queries in the finite-sum setting, i.e., $f(\mathbf{x}) = \frac{1}{n} \sum_{i=1}^{n} f_i(\mathbf{x})$.

One widely-used method for sampling from the Gibbs distribution is to use Langevin Monte Carlo (LMC) algorithm:

$$\mathbf{x}_{t+1} = \mathbf{x}_t - \eta_t \nabla f(\mathbf{x}_t) + \sqrt{2\eta_t}\epsilon_t, \tag{2}$$

where $\eta_t$ is the step size and $\epsilon_t$ is isotropic Gaussian noise. Another method that is commonly used in sampling is the Hamiltonian Monte Carlo (HMC) algorithm, which uses the principles of Hamiltonian dynamics to propose new states in a Markov Chain. It introduces the Hamiltonian

$H(\mathbf{x}, \mathbf{p}) = f(\mathbf{x}) + \frac{1}{2}\|\mathbf{p}\|^2$ with auxiliary momentum variables and updates the position ($\mathbf{x}$) and momentum ($\mathbf{p}$) by simulating Hamiltonian dynamics, which follows the equations:

$$\frac{\mathrm{d}\mathbf{x}}{\mathrm{d}t} = \frac{\partial H}{\partial \mathbf{p}}, \quad \frac{\mathrm{d}\mathbf{p}}{\mathrm{d}t} = -\frac{\partial H}{\partial \mathbf{x}}. \tag{3}$$

Similar to LMC, HMC is simulated in practice by discretizing Eq. (3). The algorithm also refreshes the momentum periodically from a random distribution, making the algorithm non-deterministic. Although effective, the computational cost of each iteration in these algorithms becomes prohibitive when the computation of the gradient is costly, such as in the finite-sum setting. To alleviate the computational burden, stochastic gradient-based samplers such as Stochastic Gradient Langevin Dynamics (SGLD) Welling & Teh (2011) and Stochastic Hamiltonian Monte Carlo (SG-HMC) Chen et al. (2014) have been proposed. Instead of computing the full gradient, these algorithms use stochastic approximation to the gradient. For example, the stochastic update for LMC becomes

$$\mathbf{x}_{t+1} = \mathbf{x}_t - \eta_t \mathbf{g}_t + \sqrt{2\eta_t}\epsilon_t. \tag{4}$$

In the finite-sum form, $\mathbf{g}_t$ can be obtained by randomly sampling a component $i \in [n]$ and computing $\nabla f_i(\mathbf{x}_t)$. While stochastic gradient methods reduce computation at each iteration, they introduce variance into the gradient estimates, which can degrade the quality of the samples and slow down convergence.

Quantum computing offers a new way to address this bottleneck. By leveraging quantum primitives such as *quantum mean estimation*, it is possible to estimate averages of stochastic gradients with *quadratically fewer* oracle calls, thereby reducing the variance cost inherent in stochastic-gradient methods. Importantly, this quantum advantage can be realized within the same oracle framework used classically. Specifically, we consider quantum oracle access of the form

$$O_{\nabla f} |\mathbf{x}\rangle |i\rangle |0\rangle \mapsto |\mathbf{x}\rangle |i\rangle |\nabla f_i(\mathbf{x})\rangle. \tag{5}$$

This oracle can be implemented by making the classical gradient oracle reversible with additional registers, so its cost is comparable to the classical setting. While computing an exact gradient still requires $O(n)$ oracle calls, our algorithms demonstrate how quantum primitives can asymptotically reduce the number of gradient queries compared to the best classical methods.

With this framework, we design quantum sampling algorithms that parallel the structure of classical LMC and HMC. It is worth noting that a straightforward replacement of the stochastic gradient step in state-of-the-art classical algorithms with a quantum variance-reduction primitive does not yield asymptotic speedups, for the following reasons:

1. The state-of-the-art classical algorithms already implement variance reduction techniques such as SVRG, SARAH, SAGA, which require full gradient computations once in a while so that the variance of the stochastic gradients can be controlled more efficiently without computing the full gradient at every iteration. Therefore, directly replacing the classical stochastic gradient estimation with quantum algorithmic primitives is not meaningful unless the full gradient computations can be done less frequently as well. In fact, using the existing classical algorithms' choice of parameters would in fact imply no speedup.

2. Quantum mean estimation algorithm requires the target variance as an input parameter, which depends on the variance of the random variable. In our setting, this corresponds to the variance of stochastic gradients at the current iteration. However, this variance depends on the trajectory of the iterates and is not bounded by a fixed constant. On the contrary, this restriction does not affect classical algorithms, as one can use a fixed batch size $b$ and then analyze the convergence.

3. Even if the current variance of the stochastic gradients is given, it is not clear how to set the target variance level. Smaller target levels increase the cost of stochastic gradient estimation, whereas larger target levels do not give any quantum speedups. This is because in this regime, one does not effectively exploit the subroutine: the algorithm essentially becomes regular LMC or HMC with no variance reduction.

To address these challenges, we develop a variance upper bound that quantifies stochastic gradient variance along the LMC trajectory. This enables us to optimize the trade-off between full-gradient computation and quantum mean estimation. Then, we optimize the target variance level in such a

Table 1: Summary of the results (some of the previous results use a different scaling of $f$ and we convert the results to the same scaling as ours in the table). Here, we mainly focus on $n$ and $\epsilon$ dependency. See Theorems 4.3, 4.4 and 4.9 for explicit dependencies on $L, \mu, \alpha, d$.

| Algorithm | Assumptions | Metric | Gradient Complexity |
|---|---|---|---|
| SG-HMC Zou & Gu (2021) | Strongly Convex | $W_2$ | $\tilde{\mathcal{O}}(n\epsilon^{-2})$ |
| SVRG-HMC Zou & Gu (2021) | Strongly Convex | $W_2$ | $\tilde{\mathcal{O}}(n^{2/3}\epsilon^{-2/3} + \epsilon^{-1})$ |
| SAGA-HMC Zou & Gu (2021) | Strongly Convex | $W_2$ | $\tilde{\mathcal{O}}(n^{2/3}\epsilon^{-2/3} + \epsilon^{-1})$ |
| CV-HMC Zou & Gu (2021) | Strongly Convex | $W_2$ | $\tilde{\mathcal{O}}(\epsilon^{-2})$ |
| SRVR-HMC Zou et al. (2019) | Dissipative Gradients | $W_2$ | $\tilde{\mathcal{O}}(n + n^{1/2}\epsilon^{-2} + \epsilon^{-4})$ |
| SVRG-LMC Kinoshita & Suzuki (2022) | LSI | KL | $\tilde{\mathcal{O}}(n + n^{1/2}\epsilon^{-1})$ |
| SARAH-LMC Kinoshita & Suzuki (2022) | LSI | KL | $\tilde{\mathcal{O}}(n + n^{1/2}\epsilon^{-1})$ |
| QSVRG-HMC [Theorem 4.3] | Strongly Convex | $W_2$ | $\tilde{\mathcal{O}}(n^{1/2}\epsilon^{-3/4} + \epsilon^{-1})$ |
| QCV-HMC [Theorem 4.4] | Strongly Convex | $W_2$ | $\tilde{\mathcal{O}}(\epsilon^{-3/2})$ |
| QSVRG-LMC [Theorem 4.9] | LSI | KL[1] | $\tilde{\mathcal{O}}(n + n^{1/3}\epsilon^{-1})$ |

way that both the cost of mean estimation and required full gradient computations decrease at the same time. By setting these optimum parameters, we prove that our algorithms achieve asymptotically fewer oracle calls than the best-known classical methods for both strongly convex and nonconvex potentials (see Table 1). We focus on SVRG- and CV-based samplers because they seem to be the most efficient samplers for finite-sum functions in the classical setting; although extensions to other gradient based samplers are possible.

We note that our algorithms rely on fault-tolerant quantum computers to implement quantum mean estimation; since such devices are not yet available, we cannot provide empirical validation at this stage. Nevertheless, establishing the theoretical foundations is essential for clarifying the scope of possible quantum speedups in sampling.

## 2 RELATED WORK

Non-asymptotic convergence rates for SGLD and SG-HMC have been analyzed extensively by Raginsky et al. (2017); Xu et al. (2018); Zou et al. (2021); Das et al. (2023) and Chen et al. (2014); Zou & Gu (2021) respectively. In the finite sum setting, more sophisticated variance reduction techniques such as SVRG Johnson & Zhang (2013), SAGA Defazio et al. (2014), SARAH Nguyen et al. (2017), and Control Variates (CV) Baker et al. (2019) have been used to reduce the variance of stochastic gradients by leveraging the gradient information from previous iterations. Although these methods were originally introduced in the context of optimization, successive works have applied these methods to improve sampling efficiency via LMC Dubey et al. (2016); Chatterji et al. (2018); Baker et al. (2019); Kinoshita & Suzuki (2022) and HMC Zou et al. (2019); Zou & Gu (2021). In particular, Zou & Gu (2021) has incorporated various variance reduction techniques to SG-HMC and analyzed convergence in Wasserstein distance for smooth and strongly convex potentials. In the non-log-concave setting, Kinoshita & Suzuki (2022) has analyzed the convergence of SVRG-LMC and SARAH-LMC for target distributions that satisfy the Log-Sobolev inequality and applied their results to optimize structured non-convex objectives.

In the context of quantum sampling, the most of the existing results makeuse of *quantum walks*, which has been shown to provide speedups for certain Markov Chain Monte Carlo (MCMC) methods by improving the mixing time of the underlying Markov chain Szegedy (2004); Somma et al. (2007; 2008); Wocjan & Abeyesinghe (2008); Chakrabarti et al. (2023). These methods have been incorporated into various domains to improve the computation time of various tasks Magniez et al. (2007); Apers & Sarlette (2019); Childs et al. (2022); Chakrabarti et al. (2023); Li & Zhang (2024); Chakrabarti et al. (2024). For sampling from continuous distributions, Kinoshita & Suzuki (2022) used quantum walk framework to improve the gradient queries to approximately sample from strongly-convex distributions. However, a key limitation of quantum walks is that they require the Markov chain to satisfy detailed balance condition. A Markov chain on $\Omega$ with transition density

---

[1]Convergence in KL divergence implies convergence in squared TV and $W_2$ distances due to Pinsker's and Talagrand's inequalities.

matrix $P$ and stationary density $\pi$ needs to satisfy for all $\mathbf{x}, \mathbf{y} \in \Omega$, $\pi(\mathbf{x})P(\mathbf{x}, \mathbf{y}) = \pi(\mathbf{y})P(\mathbf{y}, \mathbf{x})$. Unfortunately, many commonly used sampling algorithms, such as LMC and HMC, are not reversible due to the finite discretization steps involved in their implementation. The reversibilization techniques such as Metropolis reversibilization require evaluating the function exactly to compute ratio $\pi(\mathbf{x})/\pi(\mathbf{y})$ which is expensive in finite-sum case. Additionally, implementing rejection steps causes additional overheads: if a proposed move is rejected, one must revert to the previous state. However, due to the no-cloning theorem, it is not straightforward to restore the previous quantum state. Therefore, the quantum walk operator needs nontrivial techniques when it involves Metropolis correction. We refer the reviewer to recent work on this topic Claudon et al. (2025b). Moreover, even when the Markov chain is reversible, stochastic gradients introduce randomness that disrupts the coherent evolution of the quantum walk, which is a critical component of its speedup Ozgul et al. (2024). More recently, Claudon et al. (2025a) proposed a similar technique to obtain quantum speedups for nonreversible Markov chains, using the idea of geometric reversibilization with respect to the so-called "most reversible" distribution which requires $\mathcal{O}(n)$ gradient computations in this case. Another limitation of quantum walks is that they typically offer convergence guarantees in terms of total variation distance; however, many practical sampling tasks are more concerned with metrics like Wasserstein distance or Kullback-Leibler divergence.

However, hybrid algorithms that exploit quantum computing methods as subroutines are easier to implement and do not suffer from these issues. In the context of optimization, quantum algorithms such as multi-dimensional quantum mean estimation Cornelissen et al. (2022) and quantum gradient estimation Jordan (2005); Gilyén et al. (2019) have shown promise in reducing the computational cost associated with gradient-based methods van Apeldoorn et al. (2020); Chakrabarti et al. (2020); Sidford & Zhang (2023); Zhang et al. (2024); Liu et al. (2024). These techniques are particularly well-suited for addressing challenges in large-scale and noisy settings, as they can provide more accurate gradient estimates with fewer queries. However, these methods have not been considered for sampling tasks and this paper focuses on integrating these quantum techniques to enhance the efficiency of stochastic gradient-based samplers and alleviate the computational burden inherent in classical methods.

## 2.1 Preliminaries

**Notation:** Bold symbols, such as $\mathbf{x}$ and $\mathbf{y}$, are used to represent vectors, with $\|\cdot\|$ indicating the Euclidean or operator norm depending on the context. Given two scalars $a$ and $b$, we use $a \wedge b$ to denote $\min\{a, b\}$ and use $a \vee b$ to denote $\max\{a, b\}$. The notation $\tilde{O}$ is used to suppress the polylogarithmic dependencies on $d, \epsilon, L, \mu$ and $\alpha$ that will be defined later in the text.

**Quantum computation:** Quantum computation is naturally expressed in the language of linear algebra. The *computational basis* of $\mathbb{C}^d$ is given by $\boldsymbol{e}_0, \ldots, \boldsymbol{e}_{d-1}$, where $\boldsymbol{e}_i = (0, \ldots, 1, \ldots, 0)^\top$ has a 1 in the $(i+1)^{\text{st}}$ position. In *Dirac notation*, we write $|i\rangle$ (a "ket") for $\boldsymbol{e}_i$ and $\langle i|$ (a "bra") for $\boldsymbol{e}_i^\top$.

The *tensor product* of two quantum states is their Kronecker product. If $|u\rangle \in \mathbb{C}^{d_1}$ and $|v\rangle \in \mathbb{C}^{d_2}$, then

$$|u\rangle \otimes |v\rangle = (u_0 v_0, u_0 v_1, \ldots, u_{d_1-1}v_{d_2-1})^\top \in \mathbb{C}^{d_1} \otimes \mathbb{C}^{d_2}. \tag{6}$$

The fundamental unit of quantum information is the *qubit*, a state in $\mathbb{C}^2$ of the form $a|0\rangle + b|1\rangle$ with $a, b \in \mathbb{C}$ and $|a|^2 + |b|^2 = 1$. An $n$-qubit product state takes the form $|v_1\rangle \otimes \cdots \otimes |v_n\rangle \in \mathbb{C}^{2^n}$, where each $|v_i\rangle$ is a single-qubit state. Most vectors in $\mathbb{C}^{2^n}$, however, cannot be written as product states. For brevity, we often write $|u\rangle |v\rangle$ instead of $|u\rangle \otimes |v\rangle$.

Quantum states evolve under *unitary transformations*. In the circuit model, a $k$-*qubit gate* is a unitary operator in $\mathbb{C}^{2^k}$. Two-qubit gates are *universal*: any $n$-qubit unitary can be decomposed into gates acting trivially on $n-2$ qubits and nontrivially on two qubits. The *gate complexity* of an operation is defined as the number of two-qubit gates required in its circuit implementation.

Access to information of a function or a probability distribution in quantum algorithms is provided via a *quantum oracle*. Such oracles must be reversible and allow queries on superpositions of inputs. The following definition demonstrates an oracle we use in this paper for sampling from a probability distribution.

**Definition 2.1** (Quantum Sampling Oracle). Quantum sampling oracle $O_X$ of a random variable $X \in \Omega$ is given by $O_X \left| 0 \right\rangle \left| 0 \right\rangle \mapsto \sum_{X \in \Omega} \sqrt{\Pr(X)} \left| X \right\rangle \left| \text{garbage}(X) \right\rangle$.

Here, the second register contains $\left| \text{garbage}(X) \right\rangle$, which depends on $X$. The state in the (auxiliary) garbage register is usually generated in some intermediate step of computing $X$ in the first register. It is important to note that the state in this quantum sampling oracle differs from the coherent quantum sample state, as the former is entangled and we cannot simply discard the garbage register.

Another oracle used in this paper is the *stochastic gradient oracle* as specified in Eq. (5).

Beyond simulating classical random sampling, quantum oracles enable uniquely quantum effects such as interference. These underlie key techniques like amplitude amplification (central to Grover's search Grover (1996)) and amplitude estimation, both of which rely on coherent oracle access. Similar considerations apply to the quantum gradient oracle Eq. (5). Whenever a classical oracle can be realized by a circuit, the corresponding quantum oracle can be implemented by a quantum circuit with little overhead. Thus, quantum oracles provide a natural framework for analyzing the complexity of tasks such as sampling from a probability distribution and optimization.

To sample from a distribution $p$ over $\mathbb{R}^d$, it suffices to prepare the quantum state $\sum_{\mathbf{x}} \sqrt{p(\mathbf{x})} \mathrm{d} \left| \mathbf{x} \right\rangle$ and then measure it.

**Metrics:** We use several metrics to compare probability distributions over a state space $\mathcal{X}$. Let $\pi$ and $\mu$ be two probability distributions on $\mathcal{X}$. The $p$-Wasserstein distance between $\pi$ and $\mu$ is defined as $W_p(\pi, \mu) = \left( \inf_{\gamma \in \Gamma(\pi, \mu)} \mathbb{E}_{(\mathbf{x}, \mathbf{y}) \sim \gamma} \| \mathbf{x} - \mathbf{y} \|^p \right)^{1/p}$ where $\Gamma(\pi, \mu)$ is the set of all joint distributions $\gamma(\mathbf{x}, \mathbf{y})$ whose marginals are $\pi$ and $\mu$. The KL divergence of $\pi$ with respect to $\mu$ is defined as $\mathrm{KL}(\pi \| \mu) = \int_{\mathcal{X}} \mathrm{d}\mathbf{x} \pi(\mathbf{x}) \log \left( \frac{\pi(\mathbf{x})}{\mu(\mathbf{x})} \right)$ and the relative Fisher information is $\mathrm{FI}(\pi \| \mu) = \int_{\mathcal{X}} \mathrm{d}\mathbf{x} \pi(\mathbf{x}) \left\| \nabla \log \left( \frac{\pi(\mathbf{x})}{\mu(\mathbf{x})} \right) \right\|^2$. The total variation distance is defined as $\mathrm{TV}(\pi, \mu) = \sup_{A \subseteq \mathcal{X}} |\pi(A) - \mu(A)| = \frac{1}{2} \int_{\mathcal{X}} \mathrm{d}\mathbf{x} |\pi(\mathbf{x}) - \mu(\mathbf{x})|$.

In the next section, we analyze the trade-off between the error due to stochastic gradients and discretization to quantify how much quantum mean estimation techniques can provide speedups when combined with classical variance reduction methods such as SVRG and CV.

## 3 BACKGROUND

In this section, we give background on some classical and quantum algorithms for various tasks that are repeatedly referred in the main text.

### 3.1 OVERVIEW OF CLASSICAL SAMPLING ALGORITHMS

One widely-used method for sampling from the Gibbs distribution is through the Langevin diffusion equation, which follows the solution to the following stochastic differential equation (SDE):

$$d\mathbf{x}_t = -\nabla f(\mathbf{x}_t)\mathrm{d}t + \sqrt{2}\mathrm{d}\mathbf{B}_t, \tag{7}$$

where $\mathbf{B}_t$ is the standard Brownian motion. The Euler-Maruyama discretization of this SDE results in the well-known Langevin Monte Carlo (LMC) algorithm:

$$\mathbf{x}_{t+1} = \mathbf{x}_t - \eta_t \nabla f(\mathbf{x}_t) + \sqrt{2\eta_t} \epsilon_t, \tag{8}$$

In stochastic setting, once replace $\nabla f(\mathbf{x}_t)$ by stochastic gradients $\mathbf{g}(\mathbf{x}_k, \boldsymbol{\xi}_k)$.

Hamiltonian Monte Carlo (HMC) is an advanced sampling technique designed to efficiently explore high-dimensional probability distributions by introducing auxiliary momentum variables. Given a target distribution $\pi(\mathbf{x}) \propto e^{-f(\mathbf{x})}$, HMC augments the state space with momentum variables $\mathbf{p}$ and defines the Hamiltonian $H(\mathbf{x}, \mathbf{p}) = f(\mathbf{x}) + \frac{1}{2} \|\mathbf{p}\|^2$ where $\mathbf{p} \sim \mathcal{N}(0, I)$.

HMC alternates between updating the position $\mathbf{x}$ and momentum $\mathbf{p}$ by simulating Hamiltonian dynamics Eq. (3). In practice, Hamiltonian dynamics is simulated using the leapfrog integrator, which discretizes the continuous equations of motion. The key advantage of HMC is that it allows for large, efficient moves through the parameter space by leveraging gradient information

---

**Algorithm 1** SG-LMC

**input** The stochastic gradient oracle $O_{\nabla f}$, initial point $\mathbf{x}_0$, step size $\eta$, number of steps $K$

**output** Approximate sample from $\pi \propto e^{-f(\mathbf{x})}$

    **for** $t = 0$ to $K$ **do**

        Sample $\epsilon_t \sim \mathcal{N}(0, I)$

        $\mathbf{x}_{k+1} = \mathbf{x}_k - \eta_t k \mathbf{g}(\mathbf{x}_k, \boldsymbol{\xi}_k) + \sqrt{2\eta_k}\epsilon_k$,

    **end for**

    **Return** $\mathbf{x}^K$

---

and auxiliary momentum. This reduces the correlation between successive samples, particularly in high-dimensional spaces, resulting in faster convergence compared to simple random-walk methods like the Metropolis-Hastings algorithm. In practice, Hamiltonian dynamics are simulated using the leapfrog integrator, which discretizes the continuous equations of motion.

After a series of updates, the momentum $\mathbf{p}_{k+1}$ is refreshed by sampling from $\mathcal{N}(0, I)$. This discretization ensures symplecticity, preserving volume in phase space and allowing the algorithm to make large, energy-conserving moves through the parameter space.

---

**Algorithm 2** SG-HMC

**input** The stochastic gradient oracle $O_{\nabla f}$, initial point $\mathbf{x}_0$, step size $\eta$, number of leapfrog steps $S$, number of HMC proposals $T$

**output** Approximate sample from $\pi \propto e^{-f(\mathbf{x})}$

    **for** $t = 0$ to $T$ **do**

        Sample $\mathbf{p}_{St} \sim \mathcal{N}(0, I)$

        **for** $s = 0$ to $S - 1$ **do**

          $k = St + s$

          $\mathbf{x}_{k+1} = \mathbf{x}_k + \eta \mathbf{p}_k - \frac{\eta^2}{2}\mathbf{g}(\mathbf{x}_k, \boldsymbol{\xi}_k)$

          $\mathbf{p}_{k+1} = \mathbf{p}_k - \frac{\eta}{2}\mathbf{g}(\mathbf{x}_k, \boldsymbol{\xi}_k) - \frac{\eta}{2}\mathbf{g}(\mathbf{x}_{k+1}, \boldsymbol{\xi}_{k+1/2})$

        **end for**

    **end for**

    **Return** $\mathbf{x}^T$

---

Similar to SGLD, one can replace the gradients with stochastic gradients resulting in SG-HMC (See Algorithm 2). The stochastic gradients $\mathbf{g}(\mathbf{x}, \xi)$ in Algorithm 2 can be obtained using different techniques such as mini-batch, SVRG, CV. In this case, we use quantum variance reduction techniques to compute $\mathbf{g}(\mathbf{x}, \xi)$.

## 3.2 QUANTUM MEAN ESTIMATION

Quantum mean estimation is a technique to estimate the mean of a $d$-dimensional random variable $X$ up to $\epsilon$ accuracy using $\tilde{\mathcal{O}}(d^{1/2}/\epsilon)$ queries, which is a quadratic improvement in $\epsilon$ compared to classical algorithms Cornelissen et al. (2022). Although the quantum mean estimation algorithm is biased, Sidford & Zhang (2023) developed an unbiased quantum mean estimation algorithm. Specifically, for a multi-dimensional variable with mean $\mu$ and variance $\sigma^2$, unbiased quantum mean estimation outputs an estimate $\hat{\mu}$ such that $\mathbb{E}[\hat{\mu}] = \mu$ and $\mathbb{E}[\|\hat{\mu} - \mu\|^2] \leq \hat{\sigma}^2$ using $\tilde{\mathcal{O}}(d^{1/2}\sigma/\hat{\sigma})$ queries.

The following lemma shows that the mean $\mathbb{E}[X]$ for a random variable $X$ can be computed quadratically faster than classical mean estimation with respect to oracle $O_X$.

**Lemma 3.1** (Unbiased Quantum Mean Estimation Sidford & Zhang (2023)). *For a $d$-dimensional random variable $X$ with $\text{Var}[X] \leq \sigma^2$ and some $\hat{\sigma} \geq 0$, suppose we are given access to its quantum sampling oracle $O_X$ (as in Definition 2.1). Then, there is a procedure* QuantumMeanEstimation$(O_X, \hat{\sigma})$ *that uses $\tilde{\mathcal{O}}\left(\frac{d^{1/2}\sigma}{\hat{\sigma}}\right)$ queries to $O_X$ and outputs an unbiased estimate $\hat{\mu}$ of the expectation $\mu$ satisfying $\text{Var}[\hat{\mu}] \leq \hat{\sigma}^2$.*

# 4 QUANTUM SPEEDUPS FOR FINITE-SUM SAMPLING VIA GRADIENT ORACLE

We assume access to the oracle defined in Eq. (5). The goal is to approximately sample from $\pi$ by using as few gradient computations as possible without deteriorating the convergence. To this end, we introduce the first stochastic-gradient samplers that integrate unbiased quantum mean estimation with classical variance-reduction frameworks, yielding provable improvements in gradient-query complexity for both HMC and LMC. Our algorithms are actually quite simple: we replace the stochastic gradient estimation with quantum mean estimation. However, obtaining the speedup is not as simple because to lower the total computational cost, we must decrease the cost of full gradient computations as well. Even though we do not modify the full gradient estimation part, the quantum mean estimation allows less frequent full gradient computations due to improved variance reduction.

Our analysis develops a variance control results (starting with A.3) that quantifies stochastic gradient variance along the sampling trajectory, enabling us to choose optimal balance between full-gradient computations and quantum mean estimation. Building on this tool, we establish improved complexity bounds under both strong convexity and LSI assumptions, demonstrating speedups over state-of-the-art classical algorithms. For readability, we include some of the key results in the main text and defer full technical details to the appendix due to the space limitation.

## 4.1 SAMPLING UNDER STRONG CONVEXITY VIA HAMILTONIAN MONTE CARLO

First, we consider quantum speedups for Hamiltonian Monte Carlo (HMC) algorithm using quantum variance reduction techniques.

---

**Algorithm 3** QSVRG/QCV

---

**input** $O_{\nabla f}$, current iterate $\mathbf{x}_k$, smoothness constant $L$, variance scale factor $b$, epoch length $m$.
**output** Quantum variance reduced stochastic gradient $\mathbf{g}$.

1: **QSVRG:**
2: **if** $k \mod m = 0$ **then**
3: $\quad \mathbf{g}_k = \nabla f(\mathbf{x}_k)$
4: $\quad \tilde{\mathbf{x}} = \mathbf{x}_k$
5: **else**
6: $\quad$ Define oracle $O_{\text{SVRG}}^{\mathbf{x}_k}$:

$$|0\rangle |0\rangle \mapsto \frac{1}{\sqrt{n}} \sum_{i=1}^{n} |\nabla f_i(\mathbf{x}_k) - \nabla f_i(\tilde{\mathbf{x}}) + \nabla f(\tilde{\mathbf{x}})\rangle |i\rangle$$

7: $\quad \hat{\sigma}^2 = L^2 \|\mathbf{x}_k - \tilde{\mathbf{x}}\|^2 / b^2$
8: $\quad \mathbf{g}_k = \texttt{QuantumMeanEstimation}(O_{\text{SVRG}}^{\mathbf{x}_k}, \hat{\sigma}^2)$
9: **end if**

10: **QCV:**
11: Define oracle $O_{\text{CV}}^{\mathbf{x}_k}$:

$$|0\rangle |0\rangle \mapsto \frac{1}{\sqrt{n}} \sum_{i=1}^{n} |\nabla f_i(\mathbf{x}_k) - \nabla f_i(\mathbf{x}_0) + \nabla f(\mathbf{x}_0)\rangle |i\rangle$$

12: $\hat{\sigma}^2 = L^2 \|\mathbf{x}_k - \mathbf{x}_0\|^2 / b^2$
13: $\mathbf{g}_k = \texttt{QuantumMeanEstimation}(O_{\text{CV}}^{\mathbf{x}_k}, \hat{\sigma}^2)$

14: **Return** $\mathbf{g}_k$

---

We propose to replace the gradients in HMC (See Algorithm 2 in appendix) with quantum gradients computed via Algorithm 3. Essentially Algorithm 3 combines the classical variance reduction techniques with the unbiased quantum mean estimation algorithm in Lemma 3.1 to reduce the variance further. The epoch length $m$ for QSVRG determines the period where the full gradient needs to be

computed. The parameter $b$ is the quantum analog of batch size and will be determined analytically. To establish the convergence of the new samplers, we make the following assumptions in this section.

**Assumption 4.1** (Strong Convexity). *There exists a positive constant $\mu$ such that for all $\mathbf{x}, \mathbf{y} \in \mathbb{R}^d$ it holds that*

$$f(\mathbf{x}) \geq f(\mathbf{y}) + \langle \nabla f(\mathbf{y}), \mathbf{y} - \mathbf{x} \rangle + \frac{\mu}{2} \|\mathbf{x} - \mathbf{y}\|^2. \tag{9}$$

**Assumption 4.2** (Lipschitz Stochastic Gradients). *There exists a positive constant $L$ such that for all $\mathbf{x}, \mathbf{y} \in \mathbb{R}^d$ and all functions $f_i$, $i = 1, ..., n$, it holds that*

$$\|\nabla f_i(\mathbf{x}) - \nabla f_i(\mathbf{y})\| \leq L\|\mathbf{x} - \mathbf{y}\|. \tag{10}$$

We also define the condition number $\kappa = \frac{L}{\mu}$. These assumptions are standard and used in the classical analysis of HMC Zou & Gu (2021). Next, we give the main theorem for the quantum Hamiltonian Monte Carlo algorithm implemented with `QSVRG` technique.

**Theorem 4.3** (Main Theorem for `QSVRG-HMC`). *Let $\mu_k$ be the distribution of $\mathbf{x}_k$ in `QSVRG-HMC` algorithm. Suppose that $f$ satisfies Assumptions 4.1 and 4.2. Given that the initial point $\mathbf{x}_0$ satisfies $\|\mathbf{x}_0 - \arg\min_\mathbf{x} f(\mathbf{x})\| \leq \frac{d}{\mu}$, then, for $\eta = \mathcal{O}\left(\frac{\epsilon}{L^{1/2}d^{1/2}\kappa^{3/2}}\right)$, $S = \tilde{\mathcal{O}}\left(\frac{Ld^{1/2}\kappa^{3/2}}{\epsilon}\right)$, $T = \tilde{\mathcal{O}}(1)$, $b = \mathcal{O}\left(\frac{L^{1/8}\epsilon^{1/4}n^{1/2}}{d^{1/8}\kappa^{3/8}} \vee 1\right)$, and $m = n/b$, we have*

$$W_2(\mu_{ST}, \pi) \leq \epsilon.$$

*The total query complexity to the stochastic gradient oracle is $\tilde{\mathcal{O}}\left(\frac{Ld^{1/2}\kappa^{3/2}}{\epsilon} + \frac{L^{9/8}d^{7/8}\kappa^{3/4}n^{1/2}}{\epsilon^{3/4}}\right)$.*

The following theorem is for quantum Hamiltonian Monte Carlo algorithm implemented with `QCV` technique.

**Theorem 4.4** (Main Theorem for `QCV-HMC`). *Let $\mu_k$ be the distribution of $\mathbf{x}_k$ in `QCV-HMC` algorithm. Suppose that $f$ satisfies Assumptions 4.1 and 4.2. Given that the initial point $\mathbf{x}_0$ satisfies $\|\mathbf{x}_0 - \arg\min_\mathbf{x} f(\mathbf{x})\| \leq \frac{d}{\mu}$, then, for $\eta = \mathcal{O}\left(\frac{\epsilon}{L^{1/2}d^{1/2}\kappa^{3/2}}\right)$, $S = \tilde{\mathcal{O}}\left(\frac{Ld^{1/2}\kappa^{3/2}}{\epsilon}\right)$, $T = \tilde{\mathcal{O}}(1)$, and $b = \mathcal{O}\left(\frac{d^{1/4}\kappa^{3/4}}{L^{1/4}\epsilon^{1/2}} \vee 1\right)$, we have*

$$W_2(\mu_{ST}, \pi) \leq \epsilon.$$

*The total query complexity to the stochastic gradient oracle is $\tilde{\mathcal{O}}\left(\frac{Ld^{5/4}\kappa^{9/4}}{\epsilon^{3/2}}\right)$.*

We postpone the proofs of Theorems 4.3 and 4.4 to Appendix A. The main idea in these proofs is to express the variance of stochastic gradients $\sigma$ throughout the trajectory of the HMC in terms of $b$ and the distance between current iterate and the last iterate the full gradient is computed. Then, we optimize $b$ so that per cost of quantum mean estimation $\mathcal{O}(\frac{\sigma}{\delta})$ is equal to the per iteration cost of full gradient computation $\tilde{\mathcal{O}}(n/m)$ (because full gradient is only computed once in every $m$ iteration) to exploit the quantum mean estimation without full gradient estimation dominating the cost.

Theorems 4.3 and 4.4 imply that when $n = \mathcal{O}(\epsilon^{-1/2})$ the best classical (`SVRG-HMC`) and the best quantum (`QSVRG-HMC`) algorithms have $\tilde{\mathcal{O}}(\epsilon^{-1})$ gradient complexity. On the other hand, when $n = \omega(\epsilon^{-1})$, quantum algorithms have better complexity than the best classical algorithms, where the race between `QSVRG-HMC` and `QCV-HMC` depends on how large $n$ is.

*Remark* 4.5. Both the classical algorithms in Zou & Gu (2021) and quantum algorithms in this paper assume that the starting point is $(d/\mu)$-close to the minimizer $\mathbf{x}^\star = \arg\min f(\mathbf{x})$. In case this point is not given, it can be obtained using $\mathcal{O}(n)$ iterations of SGD Baker et al. (2019).

## 4.2 SAMPLING UNDER LOG-SOBOLEV INEQUALITY VIA LANGEVIN MONTE CARLO

We use SVRG-LMC for the base algorithm in Kinoshita & Suzuki (2022) and replace the stochastic gradient calculation with unbiased quantum mean estimation. This section generalizes the strong convexity assumption with the following LSI assumption, which is common in non-logconcave sampling.

**Assumption 4.6** (Log-Sobolev Inequality). We say that $\pi$ satisfies the Log-Sobolev inequality with constant $\alpha$ if for all $\rho$, it holds that

$$\mathrm{KL}(\rho||\pi) \leq \frac{1}{2\alpha}\mathrm{FI}(\rho||\pi). \tag{11}$$

This is a sampling analog of the PL (Polyak-Łojasiewicz) condition commonly used in optimization Chewi & Stromme (2024) and standard in non-log-concave sampling literature Vempala & Wibisono (2019); Ma et al. (2019); Chewi et al. (2022); Kinoshita & Suzuki (2022). We note that LSI relaxes strong convexity in the sense that for any $\mu$ strongly convex function $f$, $\pi$ satisfies the Log-Sobolev inequality with constant $\frac{\mu}{2}$. We also note that this assumption is weaker than the dissipative gradient condition Raginsky et al. (2017); Zou et al. (2019) which is used commonly in non-log-concave sampling. We highlight the key steps in the proof idea here; full technical details appear in Appendix B. First we start by bounding the variance of the stochastic gradients along the trajectory of LMC in terms of KL divergence to the target Gibbs distribution and $b$.

**Lemma 4.7** (QSVRG-LMC Variance Lemma). *Let $k' < k$ be the last iteration where the full gradient is computed in* QSVRG-LMC *and $\sigma_k^2 = \mathbb{E}\|\mathbf{g}_k - \nabla f(\mathbf{x}_k)\|^2$. Then, for $\eta^2 \leq \frac{1}{6L^2m^2}$,*

$$\sigma_{k'+l}^2 \leq \frac{16L^4\eta^2}{\alpha} \sum_{r=1}^{l} \mathrm{KL}(\mu_{k'+r-1}||\pi) + \frac{8\eta dmL^2}{b^2}. \tag{12}$$

Then we prove the following theorem for LMC with stochastic gradients, which might be of independent interest that will be proved in appendix Appendix B. Although similar convergence results exist in literature, they only seem to apply for langevin algorithm with full gradients. The proof of this theorem uses a comparison between the true SDE and and approximate SDE with drift term set to stochastic gradients and we analyze this distance in terms of KL divergence between the distributions using the variance bound above.

**Theorem 4.8** (Convergence theorem for QSVRG-LMC). *Assume that $m \leq b^2$. Then, for $\eta \leq \frac{\alpha^2}{24L^2m}$, the iterates in* QSVRG-LMC *satisfy,*

$$\mathrm{KL}(\mu_k||\pi) \leq e^{-\alpha\eta k}\mathrm{KL}(\mu_0||\pi) + \frac{64m\eta dL^2}{\alpha b^2} + \frac{24\eta dL^2}{\alpha}. \tag{13}$$

Note that the first term corresponds to convergence of continous SDE, the second term is due to stochastic gradients and the last term is due to discretization of SDE. Using the convergence theorem, we obtain the main result for LSI.

**Theorem 4.9** (Main Theorem for QSVRG-LMC). *Let $\mu_k$ be the distribution of $\mathbf{x}_k$ in* QSVRG-LMC *algorithm. Suppose that $f$ satisfies Assumptions 4.2 and 4.6. Then for $\eta = \mathcal{O}\left(\frac{\epsilon\alpha}{dL^2} \wedge \frac{\alpha}{L^2m}\right)$, $K = \tilde{\mathcal{O}}\left(\frac{L^2\log(\mathrm{KL}(\mu_0||\pi))}{\alpha^2}\left(n^{2/3} + \frac{d}{\epsilon}\right)\right)$, $b = \tilde{\mathcal{O}}(n^{1/3})$, and $m = \tilde{\mathcal{O}}(n^{2/3})$ we have*

$$\left\{\mathrm{KL}(\mu_K||\pi), \mathrm{TV}(\mu_K, \pi)^2, \frac{\alpha}{2}\mathrm{W}_2(\mu_K, \pi)^2\right\} \leq \epsilon.$$

*The total query complexity to the stochastic gradient oracle is $\tilde{\mathcal{O}}\left(\frac{L^2\log(\mathrm{KL}(\mu_0||\pi))}{\alpha^2}\left(nd^{1/2} + \frac{d^{3/2}n^{1/3}}{\epsilon}\right)\right)$.*

The proof of Theorem 4.9 is postponed to Appendix B. The term $n^{1/3}\epsilon^{-1}$ is only possible because the cost full gradient estimation is amortized in the sense that the total cost of stochastic gradient computations $KbT$ which is equal to the cost of total full gradient estimation $nTK/m$. Hence, both costs go down thanks to quantum variance reduced gradients. We note that in classical SVRG-LMC the optimum parameters $m = b = n^{1/2}$ where $b$ corresponds to inner batch size.

Our algorithm improves the dominant term in gradient complexity from $\tilde{\mathcal{O}}(n^{1/2}\epsilon^{-1})$ to $\tilde{\mathcal{O}}(n^{1/3}\epsilon^{-1})$. It is also worth mentioning that recently Huang et al. (2024) proposed a proximal sampling algorithm that uses $\tilde{\mathcal{O}}(\sigma^2\epsilon^{-1})$ gradient queries in the LSI setting when the stochastic gradients have bounded variance $\sigma^2$. However, this assumption is different from our setting since the variance in the stochastic gradients is not uniformly bounded by a constant, but it is bounded throughout the trajectory by a function of problem parameters such as $d, b, m, L, \alpha$ (See Lemma 4.7).

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

# A  PROOFS FOR HAMILTONIAN MONTE CARLO IN STRONGLY CONVEX CASE

We start with the following result in Zou & Gu (2021) that quantifies the convergence of the stochastic gradient Hamiltonian Monte Carlo algorithm in Wasserstein distance.

**Theorem A.1** (Theorem 4.4 in Zou & Gu (2021)). *Under Assumptions 4.1 and 4.2, let $D = \|\mathbf{x}^0 - \arg\min_{\mathbf{x}}(f(\mathbf{x}))\|$ and $\mu_T$ be the distribution of the iterate $\mathbf{x}^T$, then if the step size satisfies $\eta = O(L^{1/2}\sigma^{-2}\kappa^{-1} \wedge L^{-1/2})$ and $K = 1/(4\sqrt{L}\eta)$, the output of HMC satisfies*

$$W_2(\mu_T, \pi) \leq (1 - (128\kappa)^{-1})^{\frac{T}{2}}(2D + 2d/\mu)^{1/2} + \Gamma_1\eta^{1/2} + \Gamma_2\eta, \tag{14}$$

*where $\Gamma_1^2 = O(L^{-3/2}\sigma^2\kappa^2)$ and $\Gamma_2^2 = O(\kappa^2(LD + \kappa d + L^{-1/2}\sigma^2\eta))$ where $\sigma^2 = \max_{t \leq T} \mathbb{E}\|\mathbf{g}(\mathbf{x}_k, \boldsymbol{\xi}_k) - \nabla f(\mathbf{x}_k)\|^2$ is the upper bound on the variance of the gradients in the trajectory of* SG-HMC *algorithm.*

This is a generic result that applies to any HMC algorithm under Assumptions 4.1 and 4.2 that uses stochastic gradients with variance upper bounded by $\sigma^2$. Note that we do not assume a uniform upper bound for $\sigma$ that is independent of problem parameters. Instead, the variance upper bound depends on the trajectory of the algorithm, which can be characterized using theoretical analysis.

## A.1  PROOF OF QHMC-SVRG

**Lemma A.2.** *Under Assumption 4.2, if the initial point satisfies $\|\mathbf{x}^0 - \mathbf{x}^\star\| \leq \frac{d}{\mu}$, then it holds that*

$$\mathbb{E}_i\|\mathbf{g}(\mathbf{x}_k, \xi) - \nabla f(\mathbf{x}_k)\|^2 \leq L^2\|\mathbf{x}_k - \widetilde{\mathbf{x}}\|^2, \tag{15}$$

*where $\widetilde{\mathbf{x}} = \mathbf{x}_{k' < k}$ is the last iteration the full gradient is computed.*

*Proof.* The proof simply follows from the definition of variance in the SVRG algorithm and the smoothness of each component.

$$\mathbb{E}_i\|\mathbf{g}_i(\mathbf{x}_k, \xi) - \nabla f(\mathbf{x}_k)\|^2 \leq \mathbb{E}_i\|\nabla f_i(\mathbf{x}_k) - \nabla f_i(\widetilde{\mathbf{x}}) + f(\widetilde{\mathbf{x}}) - \nabla f(\mathbf{x}_k)\|^2 \tag{16}$$

$$\leq \mathbb{E}_i\|\nabla f_i(\mathbf{x}_k) - \nabla f_i(\widetilde{\mathbf{x}})\|^2 \tag{17}$$

$$\leq L^2\|\mathbf{x}_k - \widetilde{\mathbf{x}}\|^2. \tag{18}$$

$\square$

Lemma A.2 allows us to set the target variance in quantum mean estimation to be $L^2\|\mathbf{x}_k - \widetilde{\mathbf{x}}\|/b^2$. Hence, each mean estimation call takes $\mathcal{O}(d^{1/2}b)$ gradient evaluations by Lemma 3.1. The following lemma characterizes the variance of the stochastic gradients along the trajectory of the algorithm.

**Lemma A.3** (Modified Lemma C.2 in Zou & Gu (2021)). *Let $\mathbf{g}(\mathbf{x}_k, \boldsymbol{\xi}_k)$ be the vector computed using the unbiased quantum mean estimation algorithm in* QHMC-SVRG. *Then, under Assumption 4.2,*

$$\mathbb{E}\|\mathbf{g}(\mathbf{x}_k, \boldsymbol{\xi}_k) - \nabla f(\mathbf{x}_k)\|^2 \leq \frac{768m^2L^2\eta^2\kappa d}{b^2}, \tag{19}$$

*where the expectation is over both the iterate $\mathbf{x}_k$ and the noise in quantum mean estimation $\boldsymbol{\xi}_k$.*

Next, we prove the main theorem for QSVRG-HMC.

**Theorem 4.3** (Main Theorem for QSVRG-HMC). *Let $\mu_k$ be the distribution of $\mathbf{x}_k$ in* QSVRG-HMC *algorithm. Suppose that $f$ satisfies Assumptions 4.1 and 4.2. Given that the initial point $\mathbf{x}_0$ satisfies $\|\mathbf{x}_0 - \arg\min_{\mathbf{x}} f(\mathbf{x})\| \leq \frac{d}{\mu}$, then, for $\eta = \mathcal{O}\left(\frac{\epsilon}{L^{1/2}d^{1/2}\kappa^{3/2}}\right)$, $S = \tilde{\mathcal{O}}\left(\frac{Ld^{1/2}\kappa^{3/2}}{\epsilon}\right)$, $T = \tilde{\mathcal{O}}(1)$, $b = \mathcal{O}\left(\frac{L^{1/8}\epsilon^{1/4}n^{1/2}}{d^{1/8}\kappa^{3/8}} \vee 1\right)$, and $m = n/b$, we have*

$$W_2(\mu_{ST}, \pi) \leq \epsilon.$$

*The total query complexity to the stochastic gradient oracle is $\tilde{\mathcal{O}}\left(\frac{Ld^{1/2}\kappa^{3/2}}{\epsilon} + \frac{L^{9/8}d^{7/8}\kappa^{3/4}n^{1/2}}{\epsilon^{3/4}}\right)$.*

*Proof.* By the choice of $\eta$ in the theorem statement and the variance upper bound in Lemma A.3, $\eta = \mathcal{O}(L^{1/2}\sigma^{-2}\kappa^{-1} \wedge L^{-1/2})$. Therefore, by Theorem A.1, for $K = \frac{1}{4\sqrt{L}\eta}$, we have

$$\mathrm{W}_2(\mu_T, \pi) \leq (1 - (128\kappa)^{-1})^{\frac{T}{2}}(2D + 2d/\mu)^{1/2} + \Gamma_1\eta^{1/2} + \Gamma_2\eta \tag{20}$$

where,

$$\Gamma_1^2 = \mathcal{O}\left(\frac{L^{1/2}m^2\kappa^3 d\eta^2}{b^2}\right), \tag{21}$$

$$\Gamma_2^2 = \mathcal{O}\left(\kappa^3 d + \frac{L^{3/2}m^2\kappa^3 d\eta^3}{b^2}\right). \tag{22}$$

We set $bm = \mathcal{O}(n)$. The first term in Eq. (20) is $\mathcal{O}(\epsilon)$ when $T = \tilde{\mathcal{O}}(\log(1/\epsilon))$. The last two terms in Eq. (14) for QHMC-SVRG become $\mathcal{O}\left(\frac{L^{1/4}d^{1/2}\kappa^{3/2}\eta^{3/2}n}{b^2} + d^{1/2}\kappa^{3/2}\eta\right)$. For $b = \mathcal{O}(d^{-1/8}\kappa^{-3/8}\epsilon^{1/4}n^{1/2}L^{1/8} \vee 1)$ and $\eta = \mathcal{O}(\epsilon\kappa^{-3/2}d^{-1/2})$, the bias term becomes $\mathcal{O}(\epsilon)$. Using Lemma 3.1, the number of gradient calculations scales as $\tilde{O}(Ld^{1/2}\kappa^{3/2}\epsilon^{-1} + L^{9/8}d^{7/8}\kappa^{3/4}\epsilon^{-3/4}n^{1/2})$. $\square$

### A.2 PROOF OF QCV-HMC

**Lemma A.4** (Modified Lemma C.4 in Zou & Gu (2021)). *Let $\mathbf{g}(\mathbf{x}_k, \boldsymbol{\xi}_k)$ be the vector computed using the unbiased quantum mean estimation algorithm in* QHMC-CV. *Then, under Assumption 4.2,*

$$\mathbb{E}\|\mathbf{g}(\mathbf{x}_k, \boldsymbol{\xi}_k) - \nabla f(\mathbf{x}_k)\|^2 \leq \frac{688Ld\kappa}{b^2},$$

*where the expectation is over both the iterate $\mathbf{x}_k$ and the noise in quantum mean estimation $\boldsymbol{\xi}_k$.*

Next we prove the main result.

**Theorem 4.4** (Main Theorem for QCV-HMC). *Let $\mu_k$ be the distribution of $\mathbf{x}_k$ in* QCV-HMC *algorithm. Suppose that $f$ satisfies Assumptions 4.1 and 4.2. Given that the initial point $\mathbf{x}_0$ satisfies $\|\mathbf{x}_0 - \arg\min_{\mathbf{x}} f(\mathbf{x})\| \leq \frac{d}{\mu}$, then, for $\eta = \mathcal{O}\left(\frac{\epsilon}{L^{1/2}d^{1/2}\kappa^{3/2}}\right)$, $S = \tilde{\mathcal{O}}\left(\frac{Ld^{1/2}\kappa^{3/2}}{\epsilon}\right)$, $T = \tilde{\mathcal{O}}(1)$, and $b = \mathcal{O}\left(\frac{d^{1/4}\kappa^{3/4}}{L^{1/4}\epsilon^{1/2}} \vee 1\right)$, we have*

$$\mathrm{W}_2(\mu_{ST}, \pi) \leq \epsilon.$$

*The total query complexity to the stochastic gradient oracle is $\tilde{\mathcal{O}}\left(\frac{Ld^{5/4}\kappa^{9/4}}{\epsilon^{3/2}}\right)$.*

*Proof.* By the choice of $\eta$ in the theorem statement and the variance upper bound in Lemma A.4, $\eta = \mathcal{O}(L^{1/2}\sigma^{-2}\kappa^{-1} \wedge L^{-1/2})$. Therefore, by Theorem A.1, for $K = \frac{1}{4\sqrt{L}\eta}$, we have

$$\mathrm{W}_2(\mu_T, \pi) \leq (1 - (128\kappa)^{-1})^{\frac{T}{2}}(2D + 2d/\mu)^{1/2} + \Gamma_1\eta^{1/2} + \Gamma_2\eta, \tag{23}$$

where,

$$\Gamma_1 = \mathcal{O}\left(\frac{L^{-1/2}\kappa^3 d}{b^2}\right), \tag{24}$$

$$\Gamma_2 = \mathcal{O}\left(\kappa^3 d\right). \tag{25}$$

The first term in Eq. (23) is $\mathcal{O}(\epsilon)$ when $T = \tilde{\mathcal{O}}(1)$. The last two terms in Eq. (23) for QHMC-CV become $\mathcal{O}\left(\frac{L^{-1/4}d^{1/2}\kappa^{3/2}\eta^{1/2}}{b^2} + d^{1/2}\kappa^{3/2}\eta\right)$. For $b = \mathcal{O}(L^{-1/4}d^{1/4}\kappa^{3/4}\epsilon^{-1/2} \vee 1)$ and $\eta = \mathcal{O}(\epsilon d^{-1/2}\kappa^{-3/2})$, the bias term becomes $\mathcal{O}(\epsilon)$. Using Lemma 3.1, the number of gradient calculations scales as $\tilde{O}(Ld^{1/2}\kappa^{3/2}\epsilon^{-1} + L^{3/4}d^{5/4}\kappa^{9/4}\epsilon^{-3/2}) = \tilde{O}(Ld^{5/4}\kappa^{9/4}\epsilon^{-3/2})$. $\square$

## B    PROOFS FOR LSI CASE

**Lemma B.1** (Stochastic-LMC One Step Convergence). *Let $\mu_k$ be the distribution of the iterate $\mathbf{x}_k$, then if the step size satisfies $\eta = \frac{2}{3\alpha}$,*

$$\mathrm{KL}(\mu_{k+1}||\pi) \leq e^{-3\alpha\eta/2}\left[\left(1 + \frac{32\eta^3 L^4}{\alpha}\right)\mathrm{KL}(\mu_k||\pi) + 6\eta\sigma_k^2 + 16\eta^2 dL^2\right], \tag{26}$$

*where $\sigma_k^2 = \mathbb{E}_{\mathbf{x}_k, \boldsymbol{\xi}_k}\|\mathbf{g}(\mathbf{x}_k, \boldsymbol{\xi}_k) - \nabla f(\mathbf{x}_k)\|^2$.*

*Proof.* We compare one step of LMC starting at $\mathbf{x}_k$ with stochastic gradients $\mathbf{g}(\mathbf{x}_k, \boldsymbol{\xi}_k)$ to the output of continuous Langevin SDE (Eq. (7)) starting at $\mathbf{x}_k$ with true gradient $\nabla f(\mathbf{x}_t)$ after time $\eta$. This technique has been used to establish the convergence of unadjusted Langevin algorithm with full gradients under isoperimetry by Vempala & Wibisono (2019). We extend the analysis by Vempala & Wibisono (2019) to the stochastic gradient LMC. Assume that the initial point $\mathbf{x}_k$ and $\mathbf{g}(\mathbf{x}_k, \boldsymbol{\xi}_k)$ obey the joint distribution $\mu_0$. The randomness on $\mathbf{g}(\mathbf{x}_k, \boldsymbol{\xi}_k)$ depends both on the randomness on $\mathbf{x}_k$ and the randomness in the quantum mean estimation algorithm. Then, one step update of LMC algorithm with stochastic gradient yields,

$$\mathbf{x}_{k+1} = \mathbf{x}_k - \eta\mathbf{g}(\mathbf{x}_k, \boldsymbol{\xi}_k) + \sqrt{2\eta}\boldsymbol{\epsilon}_k.$$

Alternatively, $\mathbf{x}_{k+1}$ can be written as the solution of the following SDE at time $t = \eta$,

$$d\mathbf{x}_t = -\mathbf{g}_k dt + \sqrt{2}d\boldsymbol{W}_t$$

where $\mathbf{g}_k = \mathbf{g}(\mathbf{x}_k, \boldsymbol{\xi}_k)$ and $\boldsymbol{W}_t$ is the standard Brownian motion starting at $\boldsymbol{W}_0 = 0$. Let $\mu_t(\mathbf{x}_k, \mathbf{g}_k, \mathbf{x}_t)$ be the joint distribution of $\mathbf{x}_k$, $\mathbf{g}_k$, and $\mathbf{x}_t$ at time $t$. Each expectation in the proof is over this joint distribution unless specified otherwise.

Consider the following stochastic differential equation

$$d\boldsymbol{X} = \boldsymbol{v}(\boldsymbol{X})dt + \sqrt{2}d\boldsymbol{W},$$

where $\boldsymbol{v}$ is a smooth vector field and $\boldsymbol{W}$ is the Brownian motion with $\boldsymbol{W}_0 = 0$. The Fokker-Planck equation describes the evolution of probability density function $\mu_t$ as follows:

$$\frac{\partial \mu_t}{\partial t} = -\nabla \cdot (\mu_t \boldsymbol{v}) + \Delta\mu_t, \tag{27}$$

where $\nabla\cdot$ is the divergence operator and $\Delta$ is the Laplacian. Then, the Fokker Planck equation gives the following evolution for the marginal density $\mu_t(\mathbf{x}|\mathbf{x}_k, \mathbf{g}_k) = \mu_t(\mathbf{x}_t = \mathbf{x}|\mathbf{x}_k, \mathbf{g}_k)$,

$$\frac{\partial \mu_t(\mathbf{x}|\mathbf{x}_k, \mathbf{g}_k)}{\partial t} = \nabla \cdot (\mu_t(\mathbf{x}|\mathbf{x}_k, \mathbf{g}_k)\mathbf{g}_k) + \Delta\mu_t(\mathbf{x}|\mathbf{x}_k, \mathbf{g}_k). \tag{28}$$

Taking the expectation over both sides with respect to $(\mathbf{x}_k, \mathbf{g}_k) \sim \mu_0$,

$$\frac{\partial \mu_t(\mathbf{x})}{\partial t} = \mathbb{E}_{(\mathbf{x}_k, \mathbf{g}_k) \sim \mu_0}[\nabla \cdot (\mu_t(\mathbf{x}|\mathbf{x}_k)\mathbf{g}_k)] + \mathbb{E}_{(\mathbf{x}_k, \mathbf{g}_k) \sim \mu_0}[\Delta\mu_t(\mathbf{x}|\mathbf{x}_k)] \tag{29}$$

$$= \int_{\mathbb{R}^d} \nabla \cdot (\mu_t(\mathbf{x}|\mathbf{x}_k, \mathbf{g}_k)\mathbf{g}_k)\mu_0(\mathbf{x}_k, \mathbf{g}_k)d\mathbf{x}_k d\mathbf{g}_k + \int_{\mathbb{R}^d} \Delta\mu_t(\mathbf{x}|\mathbf{x}_k, \mathbf{g}_k)\mu_0(\mathbf{x}_k, \mathbf{g}_k)d\mathbf{x}_k d\mathbf{g}_k$$

$$\tag{30}$$

$$= \int_{\mathbb{R}^d} \nabla \cdot (\mu_t(\mathbf{x})\mu(\mathbf{x}_k, \mathbf{g}_k|\mathbf{x}_t = \mathbf{x})\mathbf{g}_k)d\mathbf{x}_k d\mathbf{g}_k + \Delta\mu_t(\mathbf{x}) \tag{31}$$

$$= \nabla \cdot \left(\mu_t(\mathbf{x})\mathbb{E}[\mathbf{g}_k - \nabla f(\mathbf{x}_k)|\mathbf{x}_t = \mathbf{x}] + \mu_t(\mathbf{x})\nabla\log\left(\frac{\mu_t(\mathbf{x})}{\pi(\mathbf{x})}\right)\right). \tag{32}$$

Consider the time derivative of KL divergence between $\mu_t$ and $\pi$,

$$\frac{d}{dt}\mathrm{KL}(\mu_t||\pi) = \frac{d}{dt}\int_{\mathbb{R}^d}\mu_t(\mathbf{x})\log\left(\frac{\mu_t(\mathbf{x})}{\pi(\mathbf{x})}\right)d\mathbf{x} \tag{33}$$

$$= \int_{\mathbb{R}^d}\frac{\partial\mu_t(\mathbf{x})}{\partial t}\log\left(\frac{\mu_t(\mathbf{x})}{\pi(\mathbf{x})}\right)d\mathbf{x}_t + \int_{\mathbb{R}^d}\mu_t(\mathbf{x})\frac{\partial}{\partial t}\log\left(\frac{\mu_t(\mathbf{x})}{\pi(\mathbf{x})}\right)d\mathbf{x} \tag{34}$$

$$= \int_{\mathbb{R}^d}\frac{\partial\mu_t(\mathbf{x})}{\partial t}\log\left(\frac{\mu_t(\mathbf{x})}{\pi(\mathbf{x})}\right)d\mathbf{x}_t + \int_{\mathbb{R}^d}\frac{\partial\mu_t(\mathbf{x})}{\partial t}d\mathbf{x} \tag{35}$$

$$= \int_{\mathbb{R}^d}\frac{\partial\mu_t(\mathbf{x})}{\partial t}\log\left(\frac{\mu_t(\mathbf{x})}{\pi(\mathbf{x})}\right)d\mathbf{x}_t. \tag{36}$$

The last term in the third equality vanishes since the $\mu_t$ is probability distribution and its $L_1$ norm is always 1. Then the KL divergence evolves as

$$\frac{d}{dt}\mathrm{KL}(\mu_t||\pi) = \int_{\mathbb{R}^d}\nabla\cdot\left(\mu_t(\mathbf{x})\mathbb{E}[\mathbf{g}_k - \nabla f(\mathbf{x})|\mathbf{x}_t = \mathbf{x}] + \mu_t(\mathbf{x})\nabla\log\left(\frac{\mu_t(\mathbf{x})}{\pi(\mathbf{x})}\right)\right)\log\left(\frac{\mu_t(x)}{\pi(\mathbf{x})}\right)d\mathbf{x} \tag{37}$$

$$= -\int_{\mathbb{R}^d}\mu_t(\mathbf{x})\left\langle\mathbb{E}[\mathbf{g}_k - \nabla f(\mathbf{x})|\mathbf{x}_t = \mathbf{x}] + \nabla\log\left(\frac{\mu_t(\mathbf{x})}{\pi(\mathbf{x})}\right), \nabla\log\left(\frac{\mu_t(\mathbf{x})}{\pi(\mathbf{x})}\right)\right\rangle d\mathbf{x} \tag{38}$$

$$= -\int_{\mathbb{R}^d}\mu_t(\mathbf{x})\left\|\nabla\log\left(\frac{\mu_t(\mathbf{x})}{\pi(\mathbf{x})}\right)\right\|^2 d\mathbf{x} + \mathbb{E}\left\langle\nabla f(\mathbf{x}_t) - \mathbf{g}_k, \nabla\log\left(\frac{\mu_t(\mathbf{x})}{\pi(\mathbf{x})}\right)\right\rangle. \tag{39}$$

The second term can be bounded as follows:

$$\mathbb{E}\left\langle\nabla f(\mathbf{x}_t) - \mathbf{g}_k, \nabla\log\left(\frac{\mu_t(\mathbf{x})}{\pi(\mathbf{x})}\right)\right\rangle \leq \mathbb{E}\left[\|\nabla f(\mathbf{x}_t) - \mathbf{g}_k\|^2 + \frac{1}{4}\left\|\nabla\log\left(\frac{\mu_t(\mathbf{x})}{\pi(\mathbf{x})}\right)\right\|^2\right] \tag{40}$$

$$= \mathbb{E}\|\nabla f(\mathbf{x}_t) - \mathbf{g}_k\|^2 + \frac{1}{4}\mathrm{FI}(\mu_t||\pi) \tag{41}$$

$$= \mathbb{E}\|\nabla f(\mathbf{x}_t) - \nabla f(\mathbf{x}_k) + \nabla f(\mathbf{x}_k) - \mathbf{g}_k\|^2 + \frac{1}{4}\mathrm{FI}(\mu_t||\pi) \tag{42}$$

$$\leq 2\mathbb{E}\|\nabla f(\mathbf{x}_t) - \nabla f(\mathbf{x}_k)\|^2 + 2\mathbb{E}_{\mu_t(\mathbf{x}_t,\mathbf{x}_k)}\|\nabla f(\mathbf{x}_k) - \mathbf{g}_k\|^2 \tag{43}$$

$$+ \frac{1}{4}\mathrm{FI}(\mu_t||\pi). \tag{44}$$

The first inequality holds since $\langle a, b\rangle \leq a^2 + \frac{b^2}{4}$. The last line follows from Young's inequality. Furthermore, using Lipschitzness of gradients of $f$, we have

$$\mathbb{E}\|\nabla f(\mathbf{x}_t) - \nabla f(\mathbf{x}_k)\|^2 \leq L^2\mathbb{E}\|\mathbf{x}_t - \mathbf{x}_k\|^2 \tag{45}$$

$$\leq L^2\mathbb{E}\| - t\mathbf{g}_k + \sqrt{2t}\boldsymbol{\epsilon}_k\|^2 \tag{46}$$

$$= t^2L^2\mathbb{E}_{\mu_0}\|\mathbf{g}_k\|^2 + 2tdL^2. \tag{47}$$

Plugging back these into the time derivative of KL divergence, we have

$$\frac{d}{dt}\mathrm{KL}(\mu_t||\pi) \leq -\frac{3}{4}\mathrm{FI}(\mu_t||\pi) + 2t^2L^2\mathbb{E}_{\mu_0}\|\mathbf{g}_k\|^2 + 2\mathbb{E}_{\mu_0}\|\nabla f(\mathbf{x}_k) - \mathbf{g}_k\|^2 + 4tdL^2 \tag{48}$$

$$\leq -\frac{3}{4}\mathrm{FI}(\mu_t||\pi) + (4t^2L^2 + 2)\mathbb{E}_{\mu_0}\|\nabla f(\mathbf{x}_k) - \mathbf{g}_k\|^2 + 4t^2L^2\mathbb{E}_{\mu_0}\|\nabla f(\mathbf{x}_k)\|^2 + 4tdL^2. \tag{49}$$

The third term can be bounded as follows: We choose an optimal coupling $\mathbf{x}_k \sim \mu_0(\mathbf{x}_k)$ and $\mathbf{x}^\star \sim \pi$ so that $\mathbb{E}\|\mathbf{x}_k - \mathbf{x}^\star\| = W_2(\mu_0, \pi)^2$, then using Young's inequality and smoothness of $f$,

$$\mathbb{E}_{\mu_0}\|\nabla f(\mathbf{x}_k)\|^2 \leq 2\mathbb{E}_{\mu_0}\|\nabla f(\mathbf{x}_k) - \nabla f(\mathbf{x}^\star)\|^2 + 2\mathbb{E}_{\mu_0}\|\nabla f(\mathbf{x}^\star)\|^2 \tag{50}$$

$$\leq 2L^2\mathbb{E}_{\mu_0}\|\mathbf{x}_k - \mathbf{x}_0\|^2 + 2\mathbb{E}_{\mu_0}\|\nabla f(\mathbf{x}^\star)\|^2 \tag{51}$$

$$\leq 2L^2 W_2(\mu_0, \pi)^2 + 2dL \tag{52}$$

$$\leq \frac{4L^2}{\alpha}\mathrm{KL}(\mu_0||\pi) + 2dL. \tag{53}$$

The last inequality follows from Talgrand's inequality. Hence for $t \leq \eta$ and $\eta \leq \frac{1}{2L}$, we have

$$\frac{d}{dt}\mathrm{KL}(\mu_t||\pi) \leq -\frac{3}{4}\mathrm{FI}(\mu_t||\pi) + (4t^2L^2 + 2)\mathbb{E}_{\mu_0}\|\nabla f(\mathbf{x}_k) - \mathbf{g}_k\|^2 + \frac{16t^2L^4}{\alpha}\mathrm{KL}(\mu_0||\pi) + 4tdL^2 + 8t^2dL^3 \tag{54}$$

$$\leq -\frac{3\alpha}{2}\mathrm{KL}(\mu_t||\pi) + (4t^2L^2 + 2)\mathbb{E}_{\mu_0}\|\nabla f(\mathbf{x}_k) - \mathbf{g}_k\|^2 + \frac{16t^2L^4}{\alpha}\mathrm{KL}(\mu_0||\pi) + 4tdL^2 + 8t^2dL^3 \tag{55}$$

$$\leq -\frac{3\alpha}{2}\mathrm{KL}(\mu_t||\pi) + 3\mathbb{E}_{\mu_0}\|\nabla f(\mathbf{x}_k) - \mathbf{g}_k\|^2 + \frac{16\eta^2L^4}{\alpha}\mathrm{KL}(\mu_0||\pi) + 8\eta dL^2 \tag{56}$$

$$\leq -\frac{3\alpha}{2}\mathrm{KL}(\mu_t||\pi) + 3\sigma_k^2 + \frac{16\eta^2L^4}{\alpha}\mathrm{KL}(\mu_0||\pi) + 8\eta dL^2. \tag{57}$$

The second inequality is due to Eq. (11). Equivalently, we can write,

$$\frac{d}{dt}(e^{3\alpha t/2}\mathrm{KL}(\mu_t||\pi)) \leq e^{3\alpha t/2}\left(3\sigma_k^2 + \frac{16\eta^2L^4}{\alpha}\mathrm{KL}(\mu_0||\pi) + 8\eta dL^2\right). \tag{58}$$

Integrating from $t = 0$ to $t = \eta$ gives,

$$e^{3\alpha\eta/2}\mathrm{KL}(\mu_\eta||\pi) - \mathrm{KL}(\mu_0||\pi) \leq 6\eta\sigma_k^2 + \frac{32\eta^3L^4}{\alpha}\mathrm{KL}(\mu_0||\pi) + 16\eta^2dL^2 \tag{59}$$

for $\eta \leq \frac{2}{3\alpha}$. Rearranging the terms,

$$\mathrm{KL}(\mu_\eta||\pi) \leq e^{-3\alpha\eta/2}\left[\left(1 + \frac{32\eta^3L^4}{\alpha}\right)\mathrm{KL}(\mu_0||\pi) + 6\eta\sigma_k^2 + 16\eta^2dL^2\right]. \tag{60}$$

Renaming $\mu_0 = \mu_k$ and $\mu_\eta = \mu_{k+1}$, we obtain the result in the statement.

$\square$

The statement in Lemma B.1 is generic and can be applied to any LMC algorithm with stochastic gradients with bounded variance on the trajectory of the algorithm. Note that this is different from assuming that the variance is uniformly upper bounded. Instead, we set inner loop and variance reduction parameters so that the variance does not explode along the trajectory of the algorithm.

## B.1 PROOF OF QSVRG-LMC

We start with the following lemma that characterizes the variance of the quantum stochastic gradients in QSVRG-LMC in terms of the distance between the current iterate and the reference point where the full gradient is computed.

**Lemma B.2.** *Let $\tilde{\mathbf{x}}$ be any iteration where* QSVRG-LMC *computes the full gradient. Then under Assumption 4.2, the quantum stochastic gradient $\mathbf{g}_k$ at $\mathbf{x}_k$ that is computed using $\tilde{\mathbf{x}}$ as a reference point in* QSVRG-LMC *satisfies*

$$\mathbb{E}[\|\mathbf{g}_k - \nabla f(\mathbf{x}_k)\|^2] \leq \frac{L^2\|\mathbf{x}_k - \tilde{\mathbf{x}}\|^2}{b^2} \tag{61}$$

*using $\tilde{O}(d^{1/2}b)$ gradient computations.*

*Proof.* Recall that SVRG computes the stochastic gradient $\tilde{\mathbf{g}}$ at $\mathbf{x}_k$ by the following.

$$\tilde{\mathbf{g}}_k = \nabla f_i(\mathbf{x}_k) - \nabla f_i(\tilde{\mathbf{x}}) + \nabla f(\tilde{\mathbf{x}}), \tag{62}$$

where $\tilde{\mathbf{x}}$ is the last iteration the full gradient is computed and $i$ is a component randomly chosen from $[n]$. Let $\sigma_k^2 = \mathbb{E}\|\tilde{\mathbf{g}}_k - \nabla f(\mathbf{x}_k)\|^2$. Then, $\sigma_k^2$ can be bounded in terms of the distance between $\mathbf{x}_k$ and $\tilde{\mathbf{x}}$.

$$\sigma_k^2 = \mathbb{E}[\|\nabla f_i(\mathbf{x}_k) - \nabla f_i(\tilde{\mathbf{x}}) + \nabla f(\tilde{\mathbf{x}}) - \nabla f(\mathbf{x}_k)\|^2] \tag{63}$$

$$= \mathbb{E}[\|\nabla f_i(\mathbf{x}_k) - \nabla f_i(\tilde{\mathbf{x}})\|^2] - (\mathbb{E}[\nabla f_i(\mathbf{x}_k) - \nabla f_i(\tilde{\mathbf{x}})])^2 \tag{64}$$

$$\leq \mathbb{E}[\|\nabla f_i(\mathbf{x}_k) - \nabla f_i(\tilde{\mathbf{x}})\|^2] \tag{65}$$

$$\leq L^2\|\mathbf{x}_k - \tilde{\mathbf{x}}\|^2, \tag{66}$$

where the equality follows from the fact that $\nabla f_i$ is an unbiased estimator for $\nabla f$ and the last line follows from Assumption 4.2. Hence, using unbiased quantum mean estimation in Lemma 3.1, we can obtain a random vector $\mathbf{g}_k$ such that,

$$\mathbb{E}\|\mathbf{g}_k - \nabla f(\mathbf{x}_k)\|^2 \leq \frac{L^2\|\mathbf{x}_k - \tilde{\mathbf{x}}\|^2}{b^2} \tag{67}$$

by using $\tilde{O}(d^{1/2}b)$ calls to the gradient oracle. $\qquad\square$

To be able to apply Lemma B.1, we need to characterize the expected upper bound on the variance of the stochastic gradients over the algorithm trajectory for SVRG.

**Lemma 4.7** (QSVRG-LMC Variance Lemma)**.** *Let $k' < k$ be the last iteration where the full gradient is computed in* QSVRG-LMC *and $\sigma_k^2 = \mathbb{E}\|\mathbf{g}_k - \nabla f(\mathbf{x}_k)\|^2$. Then, for $\eta^2 \leq \frac{1}{6L^2m^2}$,*

$$\sigma_{k'+l}^2 \leq \frac{16L^4\eta^2}{\alpha}\sum_{r=1}^{l}\mathrm{KL}(\mu_{k'+r-1}||\pi) + \frac{8\eta dmL^2}{b^2}. \tag{12}$$

*Proof.* Let $\tilde{\mathbf{x}} = \mathbf{x}_{k'}$. Then, by Lemma B.2, quantum stochastic gradient $\mathbf{g}_k$ satisfies

$$\mathbb{E}[\|\mathbf{g}_k - \nabla f(\mathbf{x}_k)\|^2] \leq \frac{L^2\mathbb{E}\|\mathbf{x}_k - \tilde{\mathbf{x}}\|^2}{b^2}. \tag{68}$$

Let $\tilde{\mathbf{x}} = \mathbf{y}_0$ and $\mathbf{x}_k = \mathbf{y}_k$, then using the update rule of Langevin Monte Carlo,

$$\mathbb{E}[\|\mathbf{x}_k - \tilde{\mathbf{x}}\|^2] = \mathbb{E}\left[\left\|\sum_{r=1}^{l}(\mathbf{y}_r - \mathbf{y}_{r-1})\right\|^2\right] = \mathbb{E}\left[\left\|\sum_{r=1}^{l}-\eta\mathbf{g}_{r-1} + \sqrt{2\eta}\boldsymbol{\epsilon}_{r-1}\right\|^2\right] \tag{69}$$

$$\leq \mathbb{E}\left[2\eta^2\left\|\sum_{r=1}^{l}\mathbf{g}_{r-1}\right\|^2 + 4\eta\left\|\sum_{r=1}^{l}\boldsymbol{\epsilon}_{r-1}\right\|^2\right] \tag{70}$$

$$\leq 2\eta^2 m\sum_{r=1}^{l}\mathbb{E}\|\mathbf{g}_{r-1}\|^2 + 4\eta\sum_{r=1}^{l}\|\boldsymbol{\epsilon}_{r-1}\|^2 \tag{71}$$

$$\leq 2\eta^2 m\sum_{r=1}^{l}\mathbb{E}\|\mathbf{g}_{r-1}\|^2 + 4\eta dm. \tag{72}$$

The first inequality is due to Young's inequality and the second inequality follows from the fact that the Gaussian noises at different iterations are independent and the fact that $l \leq m$. Defining $\sigma_{\max}^2 = \max_k \mathbb{E}\|\sigma_k\|^2$, we can write the first term on the right-hand side in terms of $\sigma_{\max}^2$,

$$\mathbb{E}[\|\mathbf{g}_r\|^2] = \mathbb{E}\|\mathbf{g}_r - \nabla f(\mathbf{x}_r) + \nabla f(\mathbf{x}_r)\|^2 \tag{73}$$

$$\leq 2\mathbb{E}\|\mathbf{g}_r - \nabla f(\mathbf{x}_r)\|^2 + 2\|\nabla f(\mathbf{x}_r)\|^2 \tag{74}$$

$$\leq 2\sigma_{\max}^2 + \frac{8L^2}{\alpha}\mathrm{KL}(\mu_r||\pi) + 4dL, \tag{75}$$

and using Eq. (68),

$$\sigma_{\max}^2 \leq \frac{4L^2m^2\eta^2\sigma_{\max}^2}{b^2} + \frac{16L^4\eta^2m}{b^2\alpha}\sum_{r=1}^{l}\text{KL}(\mu_{r-1}||\pi) + \frac{8dL^3\eta^2m^2}{b^2} + \frac{4\eta dmL^2}{b^2}. \tag{76}$$

If we set $\eta^2 \leq \frac{1}{6L^2m^2}$, we obtain

$$\sigma_{k'+l}^2 \leq \frac{32L^4\eta^2m}{b^2\alpha}\sum_{r=1}^{l}\text{KL}(\mu_{r-1}||\pi) + \frac{8\eta dmL^2}{b^2}. \tag{77}$$

$\square$

**Theorem 4.8** (Convergence theorem for QSVRG-LMC)**.** *Assume that $m \leq b^2$. Then, for $\eta \leq \frac{\alpha^2}{24L^2m}$, the iterates in* QSVRG-LMC *satisfy,*

$$\text{KL}(\mu_k||\pi) \leq e^{-\alpha\eta k}\text{KL}(\mu_0||\pi) + \frac{64m\eta dL^2}{\alpha b^2} + \frac{24\eta dL^2}{\alpha}. \tag{13}$$

*Proof.* Let $l < k$ be the last iteration the full gradient is computed. Then, using Lemmas 4.7 and B.1, we can write one step bound as follows.

$$\text{KL}(\mu_{k+1}||\pi) \leq e^{-3\alpha\eta/2}\left[\left(1 + \frac{32\eta^3L^4}{\alpha}\right)\text{KL}(\mu_k||\pi) + \frac{192m\eta^3L^4}{b^2\alpha}\sum_{r=l}^{k}\text{KL}(\mu_r||\pi) + \frac{48m\eta^2dL^2}{b^2} + 16\eta^2dL^2\right]. \tag{78}$$

First, we claim that the following inequality is true.

$$\text{KL}(\mu_{k+1}||\pi) \leq e^{-\alpha\eta k}\text{KL}(\mu_0||\pi) + \frac{48m\eta^2dL^2 + 16\eta^2dL^2b^2}{b^2(1 - e^{-\alpha\eta})}. \tag{79}$$

To prove Eq. (79), we use induction. For $k = 1$, the statement holds due to Eq. (78). That is,

$$\text{KL}(\mu_1||\pi) \leq e^{-3\alpha\eta/2}\left[\left(1 + \frac{224\eta^3L^4}{\alpha}\right)\text{KL}(\mu_0||\pi) + \frac{48m\eta^2dL^2}{b^2} + 16\eta^2dL^2\right] \tag{80}$$

$$\leq e^{-\alpha\eta}\text{KL}(\mu_0||\pi) + \frac{48m\eta^2dL^2}{b^2} + 16\eta^2dL^2 \tag{81}$$

$$\leq e^{-\alpha\eta}\text{KL}(\mu_0||\pi) + \frac{48m\eta^2dL^2 + 16\eta^2dL^2b^2}{b^2(1 - e^{-\alpha\eta})}. \tag{82}$$

The first inequality is due to the fact that $m \leq b^2$. The second inequality holds since $\left(1 + \frac{224\eta^3L^4}{\alpha}\right) \leq \left(1 + \frac{\eta\alpha}{2}\right) \leq e^{\alpha\eta/2}$ since $\eta \leq \frac{\alpha}{24L^2m}$. The third inequality follows from the fact that $1 - e^{-\alpha\eta} \leq 1$. Next, assume that the statement holds for $k - 1$, and then we prove the $k$-th

step of induction.

$$\text{KL}(\mu_k\|\pi) \le e^{-3\alpha\eta/2}\left[\left(1+\frac{32\eta^3L^4}{\alpha}\right)\text{KL}(\mu_{k-1}\|\pi) + \frac{192\eta^3L^4}{\alpha}\sum_{r=\ell}^{k-1}\text{KL}(\mu_\ell\|\pi) + \frac{48m\eta^2dL^2+16\eta^2dL^2b^2}{b^2}\right]$$

$$(83)$$

$$\le e^{-3\alpha\eta/2}\left(1+\frac{32\eta^3L^4}{\alpha}\right)\left(e^{-\alpha\eta(k-1)}\text{KL}(\mu_0\|\pi) + \frac{48m\eta^2dL^2+16\eta^2dL^2b^2}{b^2(1-e^{-\alpha\eta})}\right)$$

$$(84)$$

$$+ e^{-3\alpha\eta/2}\frac{192\eta^3L^4}{\alpha}\sum_{r=l}^{k-1}\left(e^{-\alpha\eta r}\text{KL}(\mu_0\|\pi) + \frac{48m\eta^2dL^2+16\eta^2dL^2b^2}{b^2(1-e^{-\alpha\eta})}\right) + \frac{48m\eta^2dL^2+16\eta^2dL^2b^2}{b^2}$$

$$(85)$$

$$\le e^{-3\alpha\eta/2}\left(1+\frac{32\eta^3L^4}{\alpha}\right)\left(e^{-\alpha\eta(k-1)}\text{KL}(\mu_0\|\pi) + \frac{48m\eta^2dL^2+16\eta^2dL^2b^2}{b^2(1-e^{-\alpha\eta})}\right)$$

$$(86)$$

$$+ e^{-3\alpha\eta/2}\frac{192m\eta^3L^4}{\alpha}e^{m\alpha\eta}\left(e^{-\alpha\eta(k-1)}\text{KL}(\mu_0\|\pi) + \frac{48m\eta^2dL^2+16\eta^2dL^2b^2}{b^2(1-e^{-\alpha\eta})}\right) + \frac{48m\eta^2dL^2+16\eta^2dL^2b^2}{b^2}$$

$$(87)$$

$$\le e^{-3\alpha\eta/2}\left(1+\frac{32\eta^3L^4}{\alpha}\right)\left(e^{-\alpha\eta(k-1)}\text{KL}(\mu_0\|\pi) + \frac{48m\eta^2dL^2+16\eta^2dL^2b^2}{b^2(1-e^{-\alpha\eta})}\right)$$

$$(88)$$

$$+ e^{-3\alpha\eta/2}\frac{96m\eta^3L^4}{\alpha}\left(e^{-\alpha\eta(k-1)}\text{KL}(\mu_0\|\pi) + \frac{48m\eta^2dL^2+16\eta^2dL^2b^2}{b^2(1-e^{-\alpha\eta})}\right) + \frac{48m\eta^2dL^2+16\eta^2dL^2b^2}{b^2}$$

$$(89)$$

$$\le e^{-3\alpha\eta/2}\left(1+\frac{128\eta^3L^4}{\alpha}\right)\left(e^{-\alpha\eta(k-1)}\text{KL}(\mu_0\|\pi) + \frac{48m\eta^2dL^2+16\eta^2dL^2b^2}{b^2(1-e^{-\alpha\eta})}\right) + \frac{48m\eta^2dL^2+16\eta^2dL^2b^2}{b^2}$$

$$(90)$$

$$\le e^{-\alpha\eta}\left(e^{-\alpha\eta(k-1)}\text{KL}(\mu_0\|\pi) + \frac{48m\eta^2dL^2+16\eta^2dL^2b^2}{b^2(1-e^{-\alpha\eta})}\right) + \frac{48m\eta^2dL^2+16\eta^2dL^2b^2}{b^2}$$

$$(91)$$

$$\le e^{-\alpha\eta k}\text{KL}(\mu_0\|\pi) + \frac{48m\eta^2dL^2+16\eta^2dL^2b^2}{b^2(1-e^{-\alpha\eta})}$$

$$(92)$$

$$\le e^{-\alpha\eta k}\text{KL}(\mu_0\|\pi) + \frac{64m\eta dL^2+24\eta dL^2b^2}{\alpha b^2}.$$

$$(93)$$

The first two inequalities are due to Eq. (78). The third and fourth inequality follow from the fact that $k-l \le m$ and $e^{m\alpha\eta} \le e^{\frac{\alpha^2}{8L^2}} \le e^{\frac{1}{8}} \le \frac{1}{2}$ for $\eta \le \frac{\alpha}{8mL^2}$ and the fifth inequality holds since $\left(1+\frac{128\eta^3L^4}{\alpha}\right) \le \left(1+\frac{\eta\alpha}{2}\right) \le e^{\alpha\eta/2}$ for $\eta \le \frac{\alpha}{24L^2m}$. The final inequality follows from the fact that $1-e^{-\alpha\eta} \ge \frac{3}{4}\alpha\eta$ when $\alpha\eta \le \frac{1}{4}$. This concludes the proof. $\square$

**Theorem 4.9** (Main Theorem for QSVRG-LMC). *Let $\mu_k$ be the distribution of $\mathbf{x}_k$ in QSVRG-LMC algorithm. Suppose that $f$ satisfies Assumptions 4.2 and 4.6. Then for $\eta = \mathcal{O}\left(\frac{\epsilon\alpha}{dL^2}\wedge\frac{\alpha}{L^2m}\right)$, $K = \tilde{\mathcal{O}}\left(\frac{L^2\log(\text{KL}(\mu_0\|\pi))}{\alpha^2}\left(n^{2/3}+\frac{d}{\epsilon}\right)\right)$, $b = \tilde{\mathcal{O}}(n^{1/3})$, and $m = \tilde{\mathcal{O}}(n^{2/3})$ we have*

$$\left\{\text{KL}(\mu_K\|\pi), \text{TV}(\mu_K,\pi)^2, \frac{\alpha}{2}\text{W}_2(\mu_K,\pi)^2\right\} \le \epsilon.$$

*The total query complexity to the stochastic gradient oracle is $\tilde{\mathcal{O}}\left(\frac{L^2\log(\text{KL}(\mu_0\|\pi))}{\alpha^2}\left(nd^{1/2}+\frac{d^{3/2}n^{1/3}}{\epsilon}\right)\right)$.*

*Proof.* Setting $b = \tilde{\mathcal{O}}(n^{1/3})$ and $m = \tilde{\mathcal{O}}(n^{2/3})$ and $\eta \le \frac{\epsilon\alpha}{176dL^2}$ the second term on the right hand side of Theorem 4.8 becomes smaller than $\epsilon/2$. By the step size requirement of Theorem 4.8,

we have $\eta \leq \frac{\epsilon\alpha}{176dL^2} \wedge \frac{\alpha}{24L^2m}$. The first term in Theorem 4.8 is smaller than $\epsilon/2$ when $K \leq \frac{\log(2\mathrm{KL}(\mu_0\|\pi)/\epsilon)}{\alpha\eta}$. Hence TV distance is smaller than $\epsilon$. The results for $\mathrm{W}_2$ distance and TV distance hold due to Talagrand's inequality Otto & Villani (2000) and Pinsker's inequality Tsybakov (2009) respectively. The total gradient complexity is $bK = \tilde{\mathcal{O}}\left(\frac{L^2\mathrm{KL}(\mu_0\|\pi)}{\alpha^2}\left(nd^{1/2} + \frac{d^{3/2}n^{1/3}}{\epsilon}\right)\right)$. $\qquad\square$

