# OpenReview forum: "Faster Sampling from Gibbs Distributions with Quantum Variance Reduction"
_ICLR.cc/2026/Conference — Submitted to ICLR 2026_

### Official Review · Reviewer_X1zp · 2025-10-30

**Soundness:** 2
**Presentation:** 2
**Contribution:** 1
**Rating:** 2
**Confidence:** 3

**Summary:**

This paper introduces quantum-enhanced stochastic gradient samplers that integrate unbiased quantum mean estimation with classical variance reduction techniques to accelerate approximate sampling from Gibbs distributions in finite-sum settings. The authors provide non-asymptotic analyses showing that their proposed QSVRG/QCV variants of HMC and LMC achieve improved gradient-query complexity over the current state-of-the-art classical methods under strong convexity and log-Sobolev assumptions. Overall, the work aims at establishing a theoretically grounded avenue for quantum speedups in sampling algorithms.

**Strengths:**

- The paper indeed tackles a crucial in many domains, i.e., sampling from Boltzmann-Gibbs type of distributions.
- It does a thoroughly analysis of related works and also carries forward a rigorous mathematical analysis of the proposed method.
- The paper is well structured, the related work section is very extensive and the preliminaries section exhaustive.

**Weaknesses:**

- The paper is very hard to follow at times: Many notions and concepts are mentioned in the introduction and not properly introduced. For example, the concept of oracle, is extensively mentioned in the first section without never being properly defined. This can represent a serious blocker for people without a strong background in quantum computing.
- I have the impression the paper is too focused on quantum computing. For this reason, it may be hardly accessible to the broader audience normally targeted at top tier ML conferences. I believe a specialised quantum computing conference or journals are a better venue for publishing this work.
- Again, also notation is sometimes not defined when appearing for the first time, thus making the paper not easy to follow. See for example eq (5) on page 2.
- The bottom of page 2 from line 84 onwards also appears fairly verbose and hard to parse. Furthermore, algorithms are hereby introduced with their acronym without appropriate refs and or explanations.
- Limitations are not discussed explicitly.
- Conclusions are also missing thus making the paper appearing incomplete.
- The experimental section is missing. While the authors admit that their theoretical works is based on fault-tolerant quantum devices, I still believe it is important to have a section that validates theoretical works. In this regards I wonder if the authors can simply use quantum computer simulators without any source of error to showcase the advantages of their approach in comparison to others.
- Without an appropriate empirical analysis, I find it unfair to claim any practical advantage of the proposed approach over existing classical algorithms. While theoretically there should be an advantage, the classical algorithm can be tested and ran while the proposed approach, as it relies on fault tolerant quantum device cannot be compared.
- I believe the authors should extend their work (if possible) by accounting for non fault tolerant quantum devices or error corrected qubits so that at least some sort of empirical evaluation can be carried out.

**Questions:**

See weaknesses.

**Details Of Ethics Concerns:**

Not applicable.

---

> ### Author Response · Authors · 2025-11-23
> **Response to X1zp**
>
> We thank the reviewer for taking the time to provide their feedback and for raising these concerns. While we appreciate all perspectives, this particular review appears to focus more on expectations regarding empirical evaluation of theoretical work rather than on specific issues with our contribution.
>
> Because our work requires fault-tolerant quantum computers, we are unfortunately unable to provide meaningful numerical experiments of the kind the reviewer suggests. While we could implement our algorithm and run small-scale simulations (e.g., on 10–20 qubits using a laptop or available compute nodes), such experiments would not provide useful insight: our results are asymptotic, and small-scale simulations cannot realistically demonstrate the behavior or advantages predicted by our theory. We agree that some research in quantum machine learning emphasizes near-term demonstrations, while other work (including ours) focuses on rigorously establishing theoretical speedups that typically require large-scale fault-tolerant quantum computers. Both directions contribute meaningfully to the field, and research on near-term devices is itself a specialized and largely heuristic domain. Unfortunately, many algorithms that offer provable quantum speedups, such as Grover’s and Shor’s, also cannot be meaningfully evaluated on near-term hardware.
>
> That said, we completely agree that theoretical claims should ultimately be validated empirically when the necessary hardware becomes available. However, we do not believe that the absence of such hardware today should diminish the value of theoretical results. By a similar logic, foundational theoretical work in computer science, including early work on deep learning prior to the availability of modern GPUs, would have needed to be dismissed due to the lack of immediate experimental validation.
> More broadly, we believe that the audience of top-tier ML conferences is open to diverse methodologies, including quantum computing, that may contribute to future advances in machine learning. Although our work employs quantum algorithmic tools, its central goal is to understand whether quantum methods can improve existing machine learning algorithms for sampling. Even researchers in quantum computing would likely agree that our focus is primarily on the ML side. For this reason, we believe this venue is appropriate, and similar works have been published in prior ICLR, ICML, and NeurIPS proceedings (Please see References).
>
> Below we address the reviewer’s specific questions and concerns:
>
> We would like to note that Section 2.1 already provides a brief tutorial introducing the necessary quantum computing background, including notation (e.g., Eq. (5) on page 2) and the definition of oracles. We suspect this section may have been overlooked. While we understand that the tutorial is not exhaustive, a fully detailed introduction would be out of scope for a research paper at a top-tier conference. Moreover, oracles are a standard concept in theoretical computer science and machine learning, typically understood as query-access models where internal computation details are abstracted away. For instance, a gradient oracle provides gradients at queried points without revealing additional internal structure. We are happy to clarify this further in a revision, but we believe omitting a full definition should not be viewed as a significant weakness in a theoretical work.
>
> Regarding the structure of the submission, our main results are summarized in Section 1, including Table 1. In the revised version, we will add a dedicated section discussing limitations and will improve the clarity of the paragraphs the reviewer found overly verbose.
>
> **References**
> 1. Tongyang Li and Ruizhe Zhang. Quantum speedups of optimizing approximately convex functions with applications to logarithmic regret stochastic convex bandits. **NeurIPS, 35:3152–3164, 2022**.
> 2. Zhang, Y., Zhang, C., Fang, C., Wang, L., & Li, T. (2024). Quantum Algorithms and Lower Bounds for Finite-Sum Optimization. In **ICML 2024**, PMLR 235: 60244–60270.
> 3. Ozgul, G., Li, X., Mahdavi, M., & Wang, C. (2024). Stochastic Quantum Sampling for Non-Logconcave Distributions and Estimating Partition Functions. **ICML 2024**, PMLR 235: 38953–38982.
> 4. Childs, A. M., Li, T., Liu, J.-P., Wang, C., & Zhang, R. (2022). Quantum algorithms for sampling log-concave distributions and estimating normalizing constants. **NeurIPS 2022**, 1686.
> 5. Bouland, A., Getachew, Y., Jin, Y., Sidford, A., & Tian, K. (2023). Quantum speedups for zero-sum games via improved dynamic Gibbs sampling. **ICML 2023**, 122.

---

> > ### Comment · Reviewer_X1zp · 2025-11-25
> >
> > Dear Authors,
> > Thank you for your detailed response. While I appreciate your clarifications, I still find that the level of machine learning novelty in the manuscript remains relatively limited. Combined with the absence of empirical validation — for understandable reasons — I believe the work does not yet meet the standards expected for publication in a top-tier ML venue such as ICLR.
> > Additionally, after considering the comments provided by the other reviewers, I respectfully decide to maintain my original score.

---

### Official Review · Reviewer_wgCL · 2025-10-31

**Soundness:** 2
**Presentation:** 1
**Contribution:** 2
**Rating:** 2
**Confidence:** 3

**Summary:**

This paper presents a new variance reduction approach for Hamiltonian Monte Carlo. The key idea is to replace the stochastic gradient estimation in SG-HMC via a quantum mean estimator. Since the quantum mean estimator has a quadratic improvement in $\epsilon$ compared to classical algorithms, the proposed idea can improve the gradient variance in HMC.

**Strengths:**

+ The idea of using a quantum mean error estimator in HMC is out-of-box. This idea can be transformative if it really works.
+ This paper also presents a set of theorertical proofs to show the potential advantage of this method in Section 4.
+ The introduction/tutorial about quantum computing is very intuitive, and it's easy to follow for generic machine learning researchers.

**Weaknesses:**

While the presented idea is very interesting, my main concern is that this paper draft looks so incomplete, and many key points are missing.
1.  No numerical/simulation results are provided to support the claimed benefit;
2.  No conclusion was made about the paper either
3. While this paper uses quantum mean estimator as a blackbox (which is OK), it did not provide the key details about quantum mean estimation. For instance, how a quamtum mean estimator will be implemented (algorithmically and in practical hardware)? Withoug such details, readers can hardly implement this idea.
4.  This paper didn't talk about the feasibility of implementing this idea either. Can this algorithm be impelmented using existing quantum hardware or quantum/classical hybrid processor? If not, how many quantum gates are needed in the future to enable real implementation?

**Questions:**

I have a few questions:
1. Can you explain how the quantum mean estimator will be implemented in the HMC context?
2. Can you explain how many quantum gates are needed to implement this framework? If a classical/quantum hybrid architecture is needed, how would this hybrid architecture look?
3. Measuring the results of a quantum computing framework is often very challenging in practical engineering implementation. Can you explain how you can measure the result of the quantum mean estimator for the gradient?

---

> ### Author Response · Authors · 2025-11-24
> **Response to Reviewer wgCL**
>
> We thank the reviewer for taking the time to provide their feedback and for raising these concerns. While we appreciate all perspectives, this review, like that of Reviewer X1zp, focuses more on expectations regarding the empirical performance of theoretical work rather than on specific issues with our contribution.
>
> Because our work requires fault-tolerant quantum computers, we are unfortunately unable to provide meaningful numerical experiments of the kind the reviewer suggests. While we could implement our algorithm and run small-scale simulations (e.g., on 10–20 qubits using a laptop or available compute nodes), such experiments would not provide useful insight: our results are asymptotic, and small-scale simulations cannot realistically demonstrate the behavior or advantages predicted by our theory. We agree that some research in quantum machine learning emphasizes near-term demonstrations, while other work (including ours) focuses on rigorously establishing theoretical speedups that typically require large-scale fault-tolerant quantum computers. Both directions contribute meaningfully to the field, and research on near-term devices is itself a specialized and largely heuristic domain. Unfortunately, many algorithms that offer provable quantum speedups, such as Grover’s and Shor’s, also cannot be meaningfully evaluated on near-term hardware.
>
> That said, we completely agree that theoretical claims should ultimately be validated empirically when the necessary hardware becomes available. However, we do not believe that the absence of such hardware today should diminish the value of theoretical results. By a similar logic, foundational theoretical work in computer science, including early work on deep learning prior to the availability of modern GPUs, would have needed to be dismissed due to the lack of immediate experimental validation.
>
> Below we address the reviewer’s specific questions and concerns:
>
> - Algorithm 3 on page 7 describes how the quantum mean estimation is incorporated into the HMC algorithm using different classical variance reduction techniques. The implementation of quantum mean estimation is standard and described in Section 3.2 and Corneliessen et al. Current quantum mean estimation techniques can only be implemented on fault-tolerant quantum computers, which are currently not available. The gate and qubit cost of such implementation is $\text{poly}(d,1/\epsilon)$.
>
> - As in the response to Reviewer 39VQ, our algorithms are hybrid, with only the gradient-estimation component implemented quantumly. This avoids the heavy coherence requirements of fully quantum approaches such as Childs et al. (2022) and Ozgul et al. (2023), whose quantum walk–based procedures must maintain a coherent state throughout the entire algorithm and thereby incur much larger error correction overhead. In contrast, our model is significantly lighter and often more realistic from an implementation standpoint. Within this limitation, our results are consistent with what is believed to be achievable, especially given that even classical lower bounds for the finite-sum, stochastic sampling regimes we study are not available.
>
> - It is not clear what the reviewer’s concern is related to measurements in the quantum mean estimation algorithm. Quantum mean estimation depends on the quantum amplitude estimation procedure, which is possibly the most standard algorithm in quantum computing. In the context of gradient estimation, we only treat the gradient as a d-dimensional variable and do not require additional techniques beyond multivariate mean estimation in Corneliessen et al.
>
> **References**
>
> 1. Arjan Cornelissen, Yassine Hamoudi, and Sofiene Jerbi. Near-optimal quantum algorithms for multivariate mean estimation. In Proceedings of the 54th Annual ACM SIGACT Symposium on Theory of Computing, STOC ’22. ACM, June 2022. doi: 10.1145/3519935.3520045. URL http://dx.doi.org/10.1145/3519935.3520045.

---

### Official Review · Reviewer_LTAk · 2025-10-31

**Soundness:** 3
**Presentation:** 3
**Contribution:** 3
**Rating:** 6
**Confidence:** 3

**Summary:**

The paper introduces quantum algorithms that accelerate approximate sampling from Gibbs distributions $\pi(x) \propto e^{-f(x)}$, where $ f = \frac{1}{n}\sum_i f_i $. The authors integrate quantum mean estimation with classical variance reduction techniques (e.g., SVRG, CV) to design quantum analogues of stochastic-gradient samplers such as Langevin Monte Carlo (LMC) and Hamiltonian Monte Carlo (HMC). They analyze how to optimally balance quantum variance estimation costs with occasional full-gradient computations to preserve quantum advantage. The resulting algorithms achieve provable asymptotic reductions in gradient-query complexity over the best classical methods under both strong convexity and Log-Sobolev assumptions.

**Strengths:**

- Originality: Introduces the framework combining quantum mean estimation with classical variance reduction (SVRG, CV) for stochastic-gradient-based sampling, bridging two previously separate areas—quantum optimization and classical sampling theory.
- Quality: Provides detailed theoretical analysis, including variance control lemmas and nonasymptotic convergence bounds under both strong convexity and Log-Sobolev assumptions, leading to rigorously proven quantum query speedups.
- Clarity and structure: The paper is well-organized, with clear motivation, formal assumptions, and explicit algorithmic descriptions that parallel classical counterparts (LMC/HMC).
- Significance: Demonstrates asymptotic improvements in gradient-query complexity (e.g., from $\tilde{O}(n^{1/2}\varepsilon^{-1})$ to $\tilde{O}(n^{1/3}\varepsilon^{-1})$), clarifying where quantum advantages can arise in sampling—a central task in machine learning and statistical physics.

**Weaknesses:**

- Complexity accounting: The relation between query complexity (oracle calls) and total gate complexity is not specified, making direct comparison with classical cost a little ambiguous.

- Parameter dependence: Several results require knowledge of problem constants such as the Log-Sobolev constant $\alpha$, which are typically unknown in practice.

- A bit concern of novelty: quantum mean estimation is widely used to accelerate machine learning and optimization tasks. Could the author explain more about its specific technical novelty in combining them with the sampling framework?

**Questions:**

See the weakness part, the concern of novelty.

---

> ### Author Response · Authors · 2025-11-22
> **Response to Reviewer LTAk**
>
> We thank the reviewer for the positive review, and we appreciate the constructive feedback.
>
> - We first note that our overall algorithm is hybrid: the outer loop runs classically, while the inner components rely on quantum mean estimation to obtain stochastic gradient estimates. As a result, the quantum resource cost arises only from the mean-estimation subroutine, which is known to require a polynomial number of gates and qubits in the problem dimension and accuracy. In particular, the gate and qubit complexity is $O(\text{poly}(d, 1/\epsilon))$ (Please see Cornelissen et al. 2022). We will make this explicit in the revised version, and we thank the reviewer for pointing out that this clarification would be helpful. For oracle cost, our oracle is essentially a classical oracle with superposition access. Any classical circuit that computes the gradient can be converted into such a quantum circuit with only logarithmic overhead in the qubit and gate count. Therefore, the cost of implementing our oracles is on par with the cost of implementing classical oracles.
>
> - The log-Sobolev constant plays a role analogous to the PL (Polyak–Łojasiewicz) condition in nonconvex optimization. In our analysis, the complexity depends on this constant through the choice of step size and the number of iterations used in quantum mean estimation, which functions analogously to a batch-size parameter. While this constant is typically unknown, the appropriate step size and the correct amount of averaging in the mean-estimation routine can be determined in practice via standard hyperparameter tuning and validation techniques.
>
> - Although quantum mean estimation itself is not new, incorporating it into stochastic sampling algorithms is highly nontrivial. Sampling algorithms such as LMC and HMC are inherently stochastic, and their convergence depends critically on the interplay between gradient accuracy and the noise scale of the Markov process. If the gradient error is pushed below the intrinsic stochasticity of the dynamics, further reduction provides no benefit and only increases quantum cost. Conversely, an insufficient number of iterations slows down convergence. A key contribution of our paper is to characterize the optimal noise level in the gradient estimates and to integrate quantum mean estimation in a way that respects this balance. Using too many quantum amplitude-estimation steps would increase cost unnecessarily, while using too few would degrade the mixing rate. Our analysis identifies how to apply quantum mean estimation optimally in this context.

---

### Official Review · Reviewer_39VQ · 2025-11-09

**Soundness:** 3
**Presentation:** 3
**Contribution:** 2
**Rating:** 4
**Confidence:** 4

**Summary:**

This paper presents quantum algorithms that accelerate the sampling from probability distributions $\pi \propto e^{-f}$, where $f = \frac{1}{n}\sum_i f_i$. By assuming access to individual gradients $\{\nabla f_i\}$, and leveraging quantum mean estimation techniques to existing variance reduction techniques in the classical literature, the new quantum algorithms achieve sub-quadratic speedups in key problem parameters such as dimension $n$ and accuracy $\epsilon$.

**Strengths:**

- Drawing ideas from quantum stochastic optimization methods, such as multi-dimensional quantum mean estimation and quantum gradient estimation, to improve large-scale, noisy sampling tasks.
- A non-trivial integration of quantum mean estimation to existing variance reduction techniques, leading to polynomial quantum speedups.
- Quantum speedups demonstrated for various sampling algorithms, including LMC and HMC. Three quantum speedups are identified in Table 1, two for strongly convex problems and one under the standard LSI assumptions.

**Weaknesses:**

All three reported quantum speedups are sub-quadratic:
- QSVRG-HMC: $n^{1/2}\epsilon^{-3/4}$ v.s. classical SVRG-HMC: $n^{2/3}\epsilon^{-2/3}$.
- QCV-HMC: $\epsilon^{-3/2}$ v.s. classical CV-HMC: $\epsilon^{-2}$
- QSVRG-LMC: $n^{1/3}\epsilon^{-1}$ v.s. classical SVRG-LMC: $n^{1/2}\epsilon^{-1}$.
It is noted that, to achieve the claimed speedups, fully fault-tolerant quantum computers are required. While theoretically non-trivial, such quantum speedups are of fairly limited value in practice. In fact, people commonly believe that a quadratic quantum speedup (in the asymptotic sense) may not be sufficient to yield a realistic performance gain due to the overhead of quantum error correction (QEC).

Moreover, this paper does not discuss the limitations of quantum speedups for this class of problems. If it can be established (or at least argued) that this type of stochastic sampling problem cannot be further accelerated (for example, with some query/sample lower bounds), it would be of greater impact on the field of quantum computing.

**Questions:**

1. The assumption on the stochastic gradient oracle appears to be quite strong, as it requires a superposition of individual gradients. In most practical problems, the individual gradients are only available in an incoherent superposition (i.e., a classical ensemble, such as batched SGD). Can you give some concrete scenarios where this type of quantum oracle is achievable, meaning that their quantum implementation costs are not significantly higher than implementing the classical stochastic gradient oracle?
2. My understanding is that the proposed quantum algorithms are actually hybrid quantum-classical algorithms, since the iteration steps are still performed on classical computers, while only the gradient estimation steps are replaced using quantum variance reduction. If that's the case, it should be mentioned explicitly, as this approach is quite different from a number of existing quantum sampling algorithms (e.g., [Childs et al., 2022], [Ozgul et al., 2024]) that do not perform classical iteration steps but produce a quantum state that encodes the target measure $\pi$:
3. Recently, there has been a new approach for probabilistic sampling using differential operators and QSVT: https://arxiv.org/abs/2505.05301. Is this a relevant approach for stochastic sampling? If so, how does the performance compare to the tabulated results in the paper?

---

> ### Author Response · Authors · 2025-11-21
> **Response to Reviewer 39VQ**
>
> We thank the reviewer for the feedback and for recognizing the novelty of combining quantum mean estimation with variance-reduced sampling methods.
>
> It is correct that our algorithms achieve sub-quadratic speedups. However, we should note that the problems we consider have certain local structures such as smoothness and strong-convexity, and in quantum computing a full quadratic speedup is typically obtained when the problem has no substantial structure via Grover type algorithms (quantum amplitude amplification/estimation). When the problem poses some structure, Grover type of algorithms are not optimal because the best classical algorithms use local structures rather than using brute force search. Therefore, the speedup is typically sub-quadratic and in many cases this is optimal. Please see some examples: Zhang et al. (2023), Ambainis et al. (2018), Bouland et al. (2023). For Monte Carlo methods, please see Chakrabarti et al. (2019), Ozgul et al. (2023).
>
>
>  The reviewer pointed out that a sub-quadratic speedup is typically suppressed due to error correction overheads. This is a very valid concern, and search for super-quadratic speedups is still an active research field. Unfortunately, such speedups are typically obtained by combining multiple quantum algorithms together. The justification for our sub-quadratic speedups is the same. In a machine learning  pipeline, our algorithms can be used as a submodule and our faster sampling can be used as a warm start for another quantum learning algorithm. For example, our algorithms can also be further combined with quantum counting algorithms to derive properties of the sampled random variables, which can yield additional speedup in terms of $\epsilon$.
> Moreover, our algorithms are hybrid, with only the gradient-estimation component implemented quantumly. This avoids the heavy coherence requirements of fully quantum approaches such as Childs et al. (2022) and Ozgul et al. (2023), whose quantum walk–based procedures must maintain a coherent state throughout the entire algorithm and thereby incur much larger error correction overhead. In contrast, our model is significantly lighter and often more realistic from an implementation standpoint. Within this limitation, our results are consistent with what is believed to be achievable, especially given that even classical lower bounds for the finite-sum, stochastic sampling regimes we study are not available.
>
> The reviewer can find responses to the specific questions below:
>
> 1- Concerning the stochastic gradient oracle, our assumption is equivalent to the standard classical model. Any classical circuit that computes an individual gradient can be made reversible and queried in superposition with only logarithmic overhead. Our algorithm does not assume access to a superposition of gradients; we only prepare a uniform superposition over indices $\{1, \dots, n\}$, which can be easily done by using $\log n$ single-qubit gates,  and query the gradient in superposition. This is a widely used oracle model in quantum algorithms (Zhang et al 2023, Ozgul et al 2023) and does not impose requirements beyond those of the classical setting.
>
> 2- The reviewer is also correct that our algorithm is hybrid. The iterative updates are executed classically, and the quantum routines are used only to obtain variance-reduced gradient estimates. We will make this clearer in the revision. This contrasts with approaches such as Childs et al. (2022) and Ozgul et al. (2023), which keep a fully coherent quantum state throughout the algorithm. Maintaining such coherence causes significantly more overhead, whereas our approach does not require long coherent lifetimes and additionally allows for guarantees in the Wasserstein distance. Fully coherent methods do not naturally yield Wasserstein bounds, since the analysis relies on inner products of quantum states, which connect more directly to the total variation distance.
>
> 3- Finally, the operator-based approach proposed by Leng et al. (2025) addresses a different regime. Their algorithm operates in a deterministic, zeroth-order setting and expresses speedups through isoperimetric quantities such as Poincaré constants. Our work focuses instead on finite-sum sampling with stochastic gradients, where the relevant dependencies are on the sample size $n$ and accuracy $\epsilon$. Since the algorithmic models and measures of complexity are different, the two approaches are not directly comparable.

---

> > ### Author Response · Authors · 2025-11-21
> >
> > **References**
> > 1.  Zhang, Y., Zhang, C., Fang, C., Wang, L., & Li, T. (2024). Quantum Algorithms and Lower Bounds for Finite-Sum Optimization. arXiv:2406.03006. https://arxiv.org/abs/2406.03006
> > 2. Ambainis, A., Balodis, K., Iraids, J., Kokainis, M., Prūsis, K., & Vihrovs, J. (2018). Quantum Speedups for Exponential-Time Dynamic Programming Algorithms. arXiv preprint arXiv:1807.05209.
> > 3. Bouland, A., Getachew, Y., Jin, Y., Sidford, A., & Tian, K. (2023). Quantum speedups for zero-sum games via improved dynamic Gibbs sampling. arXiv preprint arXiv:2301.03763. Retrieved from https://arxiv.org/abs/2301.03763
> > 4. Chakrabarti, S., Childs, A. M., Hung, S.-H., Li, T., Wang, C., & Wu, X. (2023). Quantum algorithm for estimating volumes of convex bodies. ACM Transactions on Quantum Computing, 4(3), 1–60. https://doi.org/10.1145/3588579

---

### Meta-Review · Area_Chair_RiHB · 2025-12-28

**Summary:**

This paper proposes a hybridized classical-quantum algorithm to sample from a Boltzmann-Gibbs distribution with a finite-sum potential function. The idea is to integrate unbiased quantum mean estimation with classical variance reduction techniques to achieve sub-quadratic speedup in terms of the gradient query complexity. The quantum advantage is demonstrated for various sampling algorithms including LMC and HMC. The paper is predominantly a good theoretical contribution for sampling, which is an important task in machine learning.

Several reviewers raised the concern that the proposed sampling algorithm only achieves sub-quadratic speedup over the classical sampling methods with variance reduction, and there is lack of numeric results (e.g., on quantum simulator) to support the claimed quantum advantage.

**Reviewer Concerns:**

In the rebuttal, the authors argued that this was due to the absence of fault-tolerant quantum hardware. While agreeing with the authors on this point, however I tend to share the reviewers’ concern that empirical validation (even on smaller scale simulation) would have been helpful to convince the practical usage of the proposed sampling algorithm with quantum mean estimation.

**Reviewer Scores:**

4 reviewers submitted their comments and scores (4/2/2/6) with confidence (4/3/3/3), with average score 3.5 and average confidence 3.25.

While this paper makes rigorous theoretical understandings of the gradient query complexity, I do not expect reviewers’ scores would increase due to the lack of empirical validation.

---

### Decision · Program_Chairs · 2026-01-26

Reject